# Efficient Latent Variable Modeling for Knowledge-Grounded Dialogue Generation

**Gunsoo Han[1]    Daejin Jo[1]    Daniel Wontae Nam[1]    Eunseop Yoon[2]    Taehwan Kwon[3]***
**Seungeun Rho[4]*    Kyoung-Woon On[1]    Chang D. Yoo[2]    Sungwoong Kim[5]†**

[1]Kakao Brain    [2]KAIST    [3]KRAFTON    [4]Georgia Institute of Technology    [5]Korea University

gunsoo.han@kakaobrain.com    swkim01@korea.ac.kr

## Abstract

Knowledge-grounded dialogue generation requires to first retrieve appropriate external knowledge based on a conversational context and then generate a response grounded on the retrieved knowledge. In general, these two sequential modules, a knowledge retriever and a response generator, have been separately trained by supervised data for each module. However, obtaining intermediate labels of the ground-truth knowledge is expensive and difficult especially in open-domain conversation. Latent variable modeling can circumvent it and enables a joint training without the knowledge supervision. In this paper, we propose an efficient algorithm for this latent variable modeling that is able to leverage a large amount of dialogue data. In specific, rather than directly training the complex retriever, we adapt a query generator with an off-the-shelf retriever, and the query generator and response generator are simultaneously trained over the latent variable of query. Moreover, we employ the evidence lower bound as a training objective and modify it to efficiently and robustly perform the joint training. Experimental results on diverse knowledge-grounded dialogue datasets show that the proposed algorithm achieves state-of-the-art performances even without the use of the annotated knowledge while maintaining the efficiency and scalability.

## 1 Introduction

Recently, knowledge-grounded dialogue generation has drawn increasing attention especially as a key ingredient for open-domain conversational agents (Dinan et al., 2018; Zhou et al., 2018b; Zhan et al., 2021; Liu et al., 2018; Zhou et al., 2018a; Lian et al., 2019; Zhao et al., 2020; Kim et al., 2020; Chen et al., 2022; kom; Shuster et al., 2022a,b; Cai et al., 2023; Wu et al., 2022; Lewis et al., 2020; Huang et al., 2021; Thulke et al., 2021; Anantha

---

* Work done at Kakao Brain
† Corresponding Author

et al., 2021). It usually obtains knowledge from external resources such as Wikipedia-based databases and web search engines since internal knowledge even in large-scale parametric language models (Brown et al., 2020; Thoppilan et al., 2022; Chowdhery et al., 2022) is incomplete or outdated, and moreover it can provide hallucinated information.

In order for a dialogue response to be grounded on such external knowledge, the conversational agent generally consists of a knowledge retriever, which retrieves knowledge corresponding to a given dialogue context, followed by a response generator that produces an informative response based on the dialogue context and the retrieved knowledge. In many previous methods, supervised learning has often been applied to independently optimize each module using the ground-truth or gold knowledge (Dinan et al., 2018; Shuster et al., 2022a,b; Glass et al., 2022; Adolphs et al., 2021a; Nogueira and Cho, 2017; Ma et al., 2020; Xie et al., 2022). However, human annotation of knowledge information is cumbersome, expensive, and often incomplete due to the existence of multiple possible knowledge candidates and the overlooking of the target response. In addition, existing automatic annotations are generally limited to a simple extractive question answering (Zhao et al., 2020; Liu et al., 2021). This difficulty in obtaining annotations of knowledge information could be severe under the open-domain conversation and hinders the use of large-scale dialogue data.

Therefore, there have been a number of recent approaches to learn the knowledge retriever and the response generator without the knowledge supervision. In specific, they have treated the retrieved knowledge, document, or passage as an unobserved latent variable and adapt latent variable modeling based on approximated marginalization (e.g. top-k) (Lewis et al., 2020; Huang et al., 2021; Cai et al., 2023; Guu et al., 2020), reinforcement learning (Zhao et al., 2020; Zhang et al., 2022; Chen et al., 2022; Wu et al., 2022) or variational methods (Zhan

et al., 2021; Paranjape et al., 2022; Lian et al., 2019; Kim et al., 2020; Chen et al., 2020; Xu et al., 2023). However, joint training of the retriever along with the generator under this latent variable modeling has some restrictions in utilizing the retriever. For example, a retriever needs to produce a differentiable prior probability for the gradient propagation through the marginalization or to be intermittently updated to rebuild the whole passage index during training (Karpukhin et al., 2020; Lewis et al., 2020; Zhang et al., 2022). This leads to limitation in facilitating complicated or large-scale retrievers and further in leveraging large-scale data. In addition, the posterior probability used in the previous variational methods has been modeled by a separated network, which grows the training complexity and also incurs the discrepancy between the training-time and test-time knowledge samplings.

To overcome such restrictions, in this paper, we propose an efficient latent variable modeling, named *ELVM*, for knowledge-grounded dialogue generation. In particular, we reduce the burden for training a whole retriever by employing a query generator followed by an off-the-shelf retriever that is fixed during training. More precisely, in tandem with the response generator, the query generator rather than the retriever is jointly optimized by latent variable modeling, in which a query is taken as a latent variable, on paired observations of dialogue contexts and responses.

To this end, we exploit the variational method using the evidence lower bound (ELBO) (Kingma and Welling, 2013; Jordan et al., 1999) and approximate the expected conditional likelihood in the ELBO by subset sampling from the prior distribution, which acts as the training objective for the response and the query generators. This approximation gets rid of an extra modeling of a surrogate posterior distribution or online posterior inference such as Markov Chain Monte Carlo (MCMC) and also reduces the training-inference discrepancy in knowledge retrieval. Moreover, we further modify the Kullback–Leibler (KL) regularization of the ELBO by constructing the approximated posterior distribution from the conditional likelihood and prior distribution and set this posterior distribution as a teacher for a distillation objective to further learn the query generator.

Experimental results show that the proposed ELVM allows to efficiently and robustly perform training without (1) the use of the annotated knowledge, (2) an explicit training of the knowledge retrieval, and (3) a complex posterior sampling. Especially, it significantly outperforms previous state-of-the-art methods for knowledge-grounded dialogue generation. In addition, the proposed posterior distillation improves the performances over the baseline that solely maximizes the expectation of the conditional likelihood.

Our main contributions can be summarized as:

- An efficient latent variable modeling is proposed in joint training of query generator and dialogue response generator without the knowledge supervision for knowledge-intensive dialogue generation.
- For realizing efficient yet robust joint training, the ELBO for the marginal likelihood is modified as the combination of the conditional likelihood objective and the posterior distillation objective, based on multiple prior samples.
- The proposed ELVM demonstrates its effectiveness in performing unsupervised joint training and even significant performance improvements over previous methods, being a new state-of-the-art on benchmark datasets.

## 2 Related Work

### 2.1 Knowledge-Grounded Dialogue Generation

Knowledge-grounded conversation has been extensively studied through recently released public datasets (Dinan et al., 2018; kom; Xu et al., 2022b; Kwiatkowski et al., 2019; Anantha et al., 2021; Moghe et al., 2018). Existing approaches for this task have mostly exploited two successive modules, the knowledge retriever and the knowledge-grounded response generator. Many previous works have used manual annotations of gold knowledge provided by some of public datasets to optimize the modules (Dinan et al., 2018; Shuster et al., 2022a,b; Glass et al., 2022; Adolphs et al., 2021a; Nogueira and Cho, 2017; Ma et al., 2020; Xie et al., 2022). Specifically, SeekeR (Shuster et al., 2022a) and BlenderBot3 (Shuster et al., 2022b) have recently proposed to build a series of modules by a single transformer with different prompts for each module, and trained its transformer on a large number of modular tasks using annotated datasets. However, this manual annotation is expensive, time-consuming, and often inaccurate, which impedes the utilization of a large-scale dialogue data that is especially necessary for open-domain conversation.

## 2.2 Unsupervised Joint Training

Recently, unsupervised training methods have been widely applied, and many of them have tries to jointly learn the modules without knowledge labels (Lewis et al., 2020; Huang et al., 2021; Zhao et al., 2020; Zhang et al., 2022; Zhan et al., 2021; Paranjape et al., 2022; Lian et al., 2019; Kim et al., 2020). For example, PLATO-KAG (Huang et al., 2021) has approximated the marginal likelihood by top-k knowledge samples while RetGen (Zhang et al., 2022) has trained to reward knowledge retrieval with the highest utilization in response generation by reinforcement learning and mixture-of-experts ensembling.

Meanwhile, latent variable modeling based on variational methods has also been developed by several recent works. CoLV (Zhan et al., 2021) has introduced collaborative latent spaces to reflect the inherent correlation between the knowledge selection and the response generation. SKT (Kim et al., 2020) has developed a sequential latent variable model for the multi-turn knowledge-grounded dialogue generation. In order to perform the posterior sampling of knowledge selection during joint training, some works have proposed to separately train the posterior distribution model (Paranjape et al., 2022; Lian et al., 2019) or the posterior information prediction model (Chen et al., 2020). Very recently, SPI (Xu et al., 2023) has applied short-run MCMC (Erik et al., 2019) for posterior sampling on the collaborative latent spaces.

While these latent variable modeling algorithms can effectively perform unsupervised joint training, the entire training of the retriever is still difficult and imposes restrictions on the selection of the retriever in terms of both a search algorithm and a knowledge resource. Furthermore, the additional posterior sampling through separate networks or multiple iterations in the previous variational methods also leads to increased training complexity as well as the training-inference discrepancy in knowledge generation. In contrast, the proposed ELVM can employ any kind of retriever in latent variable modeling since it controls a retrieval output by only changing of a generated query. Moreover, ELVM removes an extra posterior modeling or sampling by prior subset sampling and approximated posterior distillation, which leads to efficient training without the discrepancy.

## 2.3 Query Generation

When an off-the-shelf retriever is used for knowledge retrieval, how to make an input text query is important for obtaining appropriate documents. Especially, in knowledge-grounded dialogue, a self-contained query should be generated from the multi-turn dialogue context. A number of prior works have tried to train a supervised query rewriting model by human rewrites (Yu et al., 2020; Lin et al., 2020; Vakulenko et al., 2021; Voskarides et al., 2020). Similar to the knowledge annotation, the limitations of manual query annotation have came up with unsupervised learning of the query generator. A number of works have proposed a novel rewards for reinforcement learning of a query rewriter (Wu et al., 2022; Chen et al., 2022).

QKConv (Cai et al., 2023) has proposed an unsupervised query enhanced method for knowledge-grounded conversation. This work is similar to ours in that it consists of a query generator, an off-the-shelf knowledge retriever, and a response generator, and unsupervised joint training is applied to the query generator and the response generator. However, their training objective is based on the approximated marginal likelihood over candidate queries, which is different from our objective that includes the posterior distillation based on the ELBO loss.

# 3 Efficient Latent Variable Model for Knowledge-Grounded Dialogue Generation

## 3.1 Setup

In this section, we explain an overall structure of our ELVM, as described in Figure 1a, and define notations. Unlike previous works of document (passage) retrieval (Zhang et al., 2022; Huang et al., 2021; Lewis et al., 2020) using a query of an embedded vector of a dialogue context $x$, we use a text query of natural language, which enables to utilize any kind of retriever for dialogue response generation. We define the probability of generating a natural language query $u$ for a given dialogue context $x$, $p_\phi(u|x)$, as below:

$$p_\phi(u|x) = \prod_t p_\phi(u_t|u_{0:t-1}, x), \qquad (1)$$

where $u_t$ is the $t$-th token in $u$ and $p_\phi$ is the probability obtained from the model parameterized by $\phi$. Similarly, the probability of generating a response $y$ for a given query $u$ and dialogue context $x$ is defined as

$$p_\theta(y|Z(u), x) = \prod_t p_\theta(y_t|y_{0:t-1}, Z(u), x), \quad (2)$$

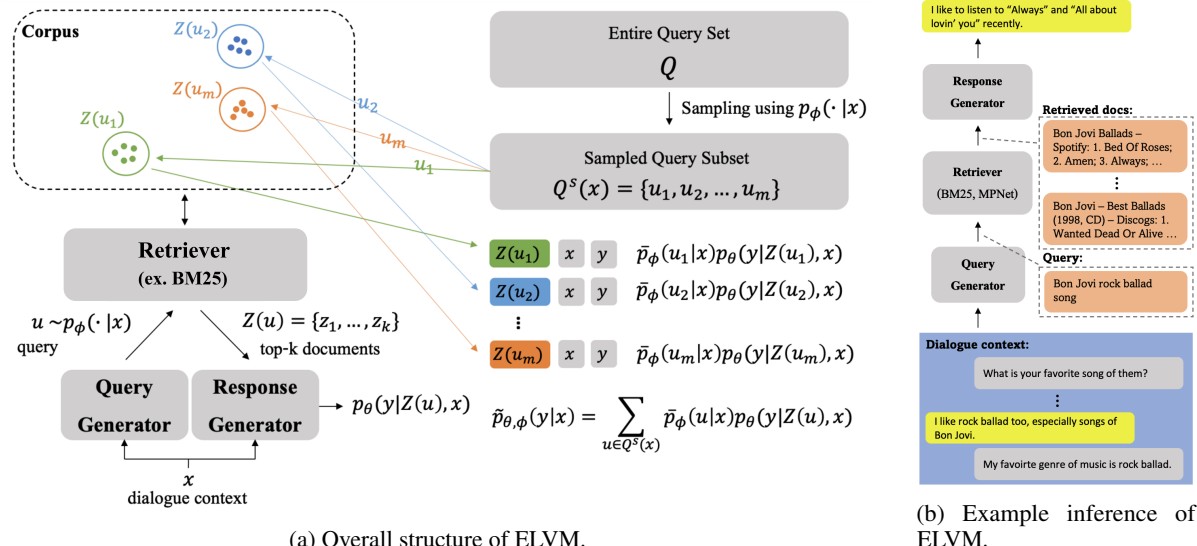

(a) Overall structure of ELVM.

(b) Example inference of ELVM.

Figure 1: Overall structure and example inference case of ELVM. During training, for a given dialogue context $x$, the query generator generates a set of unique queries $Q^s$ and for each query $u$ a set of top-$k$ documents $Z(u)$ is retrieved using the off-the-shelf retriever. Then, response generator utilizes both $x$ and $Z(u)$: (a) The approximated marginal likelihood $\tilde{p}_{\theta,\phi}(y|x)$ used during training is defined by $Q^s$ and the re-normalized prior $\bar{p}_\phi(u|x)$; (b) At inference time, a single sampled query $u$ is used for retrieving multiple documents and producing a final response.

where $y_t$ is the $t$-th token in $y$, $Z(u) = \{z_1, z_2, ..., z_k\}$ is the set of retrieved documents (passages) by $u$, and $\theta$ is the model parameters to produce the response probability. Here, we use top-$k$ result obtained by an off-the-shelf retriever such as BM25 (Robertson et al., 2009) and DPR (Karpukhin et al., 2020) and freeze the retriever. We assume that there is a deterministic mapping from $u$ to $Z$. We now set a query $u$ as a latent variable. Then, using the prior $p_\phi(u|x)$ associated with the query generator and the conditional likelihood $p_\theta(y|Z(u), x)$ corresponding to the response generator, we can define the marginal likelihood of the knowledge-grounded response $y$ given the dialogue context $x$ as below:

$$p_{\theta,\phi}(y|x) = \sum_{u \in Q} p_\phi(u|x)p_\theta(y|Z(u), x), \quad (3)$$

where $Q$ is the set of all possible queries. Since it is intractable to marginalize over all queries, we sample a small subset composed of $m$ unique queries, $Q^s(x) = \{u_1, u_2, ..., u_m\}$ using $p_\phi(\cdot|x)$ and then marginalize over this set as below:

$$\tilde{p}_{\theta,\phi}(y|x) = \sum_{u \in Q^s} \bar{p}_\phi(u|x)p_\theta(y|Z(u), x), \quad (4)$$

where

$$\bar{p}_\phi(u|x) = \frac{p_\phi(u|x)}{\sum_{u' \in Q^s} p_\phi(u'|x)}. \quad (5)$$

We construct $Q^s$ by obtaining $m$-unique queries sampled from $p_\phi(\cdot|x)$. It is noted that our response generator takes multiple retrieved documents efficiently using FiD (Izacard and Grave, 2020), which is different from the use of a single document for response generation in previous methods. In addition, while we can utilize relative matching scores from the retriever to assign varying likelihoods to each document given a query for $p(Z(u)|u)$, we have noticed that this approach can occasionally hinder the effective and comprehensive learning of the query generator. During the training process, that will be described in the next subsection, our objective is to amplify the impact of the generated query. Consequently, we opt for a deterministic mapping from a query to a set of retrieved documents as a unified whole, aiming to enhance the gradient flow.

## 3.2 Joint Training

Our training objective is the ELBO (Kingma and Welling, 2013; Jordan et al., 1999) of $\log \tilde{p}_{\theta,\phi}$ which can be written as below:

$$\log \tilde{p}_{\theta,\phi}(y|x) \geq \mathbb{E}_{u \sim q(u|x,y)}[\log p_\theta(y|Z(u), x)] - D_{KL}\big(q(u|x,y)||\bar{p}_\phi(u|x)\big), (6)$$

where $q(u|x,y)$ is the variational posterior. Note that the equality holds when $q(u|x,y)$ is identical to the true posterior distribution of it. In this work,

we do not parameterize $q(u|x,y)$ and approximate the expected conditional likelihood in the ELBO by sampling $u$ from not $q(u|x,y)$ but $\bar{p}_\phi(u|x)$ such as

$$\mathbb{E}_{u\sim q(u|x,y)}[\log p_\theta(y|Z(u),x)]$$
$$\approx \mathbb{E}_{u\sim \bar{p}_\phi(u|x)}[\log p_\theta(y|Z(u),x)]. \quad (7)$$

This approximation is due to that it minimizes the discrepancy between training and test time inference and it does not separately model $q(u|x,y)$ or complicatedly sample from it during training. Then, we obtain the following two losses corresponding to the expected conditional likelihood and the KL regularization such as

$$\mathcal{L}^{y,u}(\theta,\phi) = -\mathbb{E}_{u\sim \bar{p}_\phi(u|x)}[\log p_\theta(y|Z(u),x)], (8)$$
$$\mathcal{L}^u(\phi) = D_{KL}\big(q(u|x,y)||\bar{p}_\phi(u|x)\big). \quad (9)$$

Moreover, we define $q(u|x,y)$ in $\mathcal{L}^u(\phi)$ by the approximated posterior derived from the prior and the likelihood such as

$$q(u|x,y) = \frac{p_\theta(y|Z(u),x)\bar{p}_\phi(u|x)}{\sum_{u'\in Q^s} p_\theta(y|Z(u'),x)\bar{p}_\phi(u'|x)}. \quad (10)$$

We set this posterior distribution $q(u|x,y)$ as a teacher for a distillation to update the prior associated with the query generation at test time. Namely, $\mathcal{L}^u(\phi)$ becomes the posterior distillation loss, and our prior is explicitly updated to be similar to the posterior. Here, the gradient is not propagated from $q(u|x,y)$, and $q(u|x,y)$ can be computed easily without additional model or approximation since we define it over the subset $Q^s$.

To sum up, we obtain the conditional likelihood loss and the posterior distillation loss by multiple samples from the prior distribution that is aligned with query sampling at the inference time. We would mitigate the bias of prior sampling by multiple query samples for each input during training. In addition, our modification of ELBO is different from the expectation-maximization (EM) algorithm in that we sample a latent variable from the prior distribution rather than the posterior distribution that is used for sampling of E-step in EM. Moreover, in contrast to the M-step loss, we use the prior-weighted conditional likelihood and the gradient is also propagated to the sampled latents and the corresponding prior model. For experiments in this paper, the query generator $\phi$ and the response generator $\theta$ share parameters, with different input prompts to distinguish each module, for simplicity.

Overall, the loss for our joint training is

$$\mathcal{L}(\theta,\phi) = \mathcal{L}^{y,u}(\theta,\phi) + \beta\mathcal{L}^u(\phi), \quad (11)$$

where $\beta$ is the relative weight for the posterior distillation loss.

## 4 Experiment

### 4.1 Experimental Setup

**Dataset.** Our experiments are conducted on two widely used knowledge-grounded dialogue datasets, Wizard of Wikipedia (WoW) (Dinan et al., 2018) and QReCC (Anantha et al., 2021). WoW is structured around dialogues that are grounded in knowledge sentences sourced from Wikipedia, consisting of roughly 18K, 2K, and 2K dialogues, for training, validation, and test subsets, respectively. QReCC is a comprehensive compilation of conversational data, consisting of 14K open-domain conversations comprising 80K question-answer pairs.

| Task | Dataset | Train | Valid |
|---|---|---|---|
| **Query** | WizInt (Komeili et al., 2022) | 351,375 | 2,467 |
| | Fits (xu2) | 3,587 | 392 |
| **KG Response** | WizInt (Komeili et al., 2022) | 22,488 | 1,687 |
| | Fits (xu2) | 6,279 | 656 |
| | Ms Marco (Nguyen et al., 2017) | 281,636 | 36,859 |
| | NQ Open (Adolphs et al., 2021b) | 79,168 | 8.757 |
| **Dialogue Response** | WizInt (Komeili et al., 2022) | 8,335 | 587 |
| | MSC (Xu et al., 2022a) | 105,549 | 17,691 |
| | SaferDialogues (Ung et al., 2022) | 6,306 | 7,88 |
| | PersonaChat (Zhang et al., 2018) | 131,438 | 7.801 |
| | EmpatheicDialogues (Rashkin et al., 2019) | 64,636 | 5,738 |

Table 1: List of dataset and number of train and validation examples used for fine-tuning R2C2 model to obtain R2C2-PT. KG is an abbreviation for knowledge-grounded.

**Training.** We begin with publicly available R2C2 pre-trained transformer model (Shuster et al., 2022a) with 400M parameter size and fine-tune it on multiple datasets to acquire the essential skill sets for knowledge-grounded dialogue such as query generation and knowledge-grounded response generation. Detailed tasks and statistics are shown in Table 1. In line with Shuster et al. (2022b), we adopt a modular system with a single model that incorporates control tokens into the dialogue context to perform different modules. For example, appending the control token `__generate-query__` to the dialogue context promotes the model to generate relevant search query. We name the initial model from this pre-training procedure as R2C2-PT. Subsequently, we apply our proposed algorithm to R2C2-PT model,

| Model | Seen | | | | | Unseen | | | | |
|---|---|---|---|---|---|---|---|---|---|---|
| | PPL ↓ | B3 ↑ | B4 ↑ | R1 ↑ | R2 ↑ | PPL ↓ | B3 ↑ | B4 ↑ | R1 ↑ | R2 ↑ |
| PostKS (Lian et al., 2019) | 79.1 | - | - | 13.0 | 1.0 | 193.8 | - | - | 13.1 | 1.0 |
| ITDD (Li et al., 2019) | 17.8 | 4.0 | 2.5 | 16.2 | - | 44.8 | 2.1 | 1.1 | 11.4 | - |
| SKT (Kim et al., 2020) | 52.0 | - | - | 19.3 | 6.8 | 81.4 | - | - | 16.1 | 4.2 |
| PIPM (Chen et al., 2020) | 42.7 | - | 3.3 | 19.9 | 7.3 | 65.7 | - | 2.5 | 17.6 | 5.4 |
| ZRKGC (Li et al., 2021) | 40.4 | - | - | - | - | 41.5 | - | - | - | - |
| CoLV (Zhan et al., 2021) | 39.6 | - | 2.9 | 20.6 | 7.9 | 54.3 | - | 2.1 | 19.7 | 6.3 |
| KnowledGPT (Zhao et al., 2020) | 19.2 | 9.5 | 7.2 | 22.0 | 7.9 | 22.3 | 8.3 | 6.0 | 20.5 | 6.7 |
| SPI (Xu et al., 2023) | 17.1 | 10.2 | 7.7 | 22.7 | 8.8 | 19.1 | 9.6 | 7.3 | 22.0 | 8.5 |
| R2C2 | 40.7 | 1.7 | 0.9 | 12.3 | 1.9 | 46.6 | 1.6 | 0.9 | 11.9 | 5.4 |
| R2C2-PT | 24.7 | 6.0 | 4.4 | 17.3 | 5.2 | 30.4 | 6.2 | 4.6 | 17.7 | 5.4 |
| ELVM-OK | 19.8 | 12.0 | 9.5 | 25.3 | 10.7 | 25.8 | 12.1 | 9.7 | 25.2 | 10.5 |
| ELVM-EM-Like | 16.2 | 11.6 | 9.1 | 25.7 | 10.7 | 19.4 | 12.0 | 9.5 | 25.7 | 10.8 |
| ELVM-from-R2C2 | 15.8 | 14.5 | 11.8 | 28.8 | 13.3 | 18.9 | 14.3 | 11.5 | 28.8 | 13.3 |
| ELVM | **14.6** | **14.7** | **11.9** | **29.3** | **13.9** | **18.0** | **14.5** | **11.8** | **29.2** | **13.7** |

Table 2: Performance comparison in WoW. PPL represents perplexity, B3 and B4 denote BLEU-3 and BLEU-4 scores, respectively, and R1 and R2 indicate Rouge-1 and Rouge-2 scores. Our proposed model, ELVM, achieves new SOTA performance for both seen and unseen tasks with a substantial margin.

| Model | Seen | | | Unseen | | |
|---|---|---|---|---|---|---|
| | R@1 | R@5 | R@10 | R@1 | R@5 | R@10 |
| R2C2-PT | 25.7 | 63.8 | 74.3 | 29.2 | 64.8 | 76.7 |
| ELVM-EM-Like | 26.4 | 63.9 | 74.7 | 28.9 | 64.5 | 75.8 |
| ELVM-$\beta$=0 | 26.1 | 64.6 | 75.7 | 28.9 | 65.3 | 76.8 |
| ELVM | **29.1** | **68.4** | **80.3** | **30.4** | **68.3** | **80.3** |

Table 3: Comparison of document recall in WoW between ELVM variants. Note that we omit the performance of ELVM-OK whose performance is identical to R2C2-PT since it is trained with keeping its query generator frozen.

| Model | F1 ↑ | EM ↑ | R1 ↑ | R2 ↑ | PPL ↓ |
|---|---|---|---|---|---|
| Q.Rewriting (Raposo et al., 2022) | 18.9 | 1.0 | - | - | - |
| DPR-IHN (Kim and Kim, 2022) | 30.4 | 4.7 | - | - | - |
| QKConv (Cai et al., 2023) | 33.5 | 5.9 | - | - | - |
| R2C2-PT | 22.8 | 1.3 | 25.7 | 11.5 | 9.4 |
| ELVM-OK | 33.0 | 4.5 | 35.6 | 21.9 | 5.6 |
| ELVM | **36.5** | **6.2** | **39.0** | **25.8** | **4.8** |

Table 4: Performance comparison in QReCC with QK-Conv (Cai et al., 2023) which is the previous SOTA on this task. We report F1 scores, exact match (EM), Rouge score (R) and perplexity (PPL).

| Model | Fluency (%) | | Relevance (%) | |
|---|---|---|---|---|
| | Seen | Unseen | Seen | Unseen |
| KnowledGPT | 69.8 | 62.9 | 41.5 | 41.1 |
| ELVM | 73.5 | **77.5**[*] | 49.5 | **63.6**[**] |

Table 5: Human evaluation results on WoW test. A pairwise t-test is conducted to verify statistical significance of the improvements, and the corresponding results in bold are significantly better than those from the baseline model ([**]: $p < 0.01$, [*]: $p < 0.025$).

resulting in ELVM. Furthermore, to dispel concerns that our training gains might predominantly stem from R2C2-PT's comprehensive fine-tuning across multiple datasets rather than ELVM's inherent efficacy, we opted to utilize the base R2C2 model. Upon integrating it with ELVM, the resulting configuration is denoted as ELVM-from-R2C2.

During the training of ELVM, we employ a sampling-based generation method, specifically nucleus sampling (Holtzman et al., 2020), to promote diverse query generation in $p_\phi$. When retrieving documents based on the generated queries, we utilize an off-the-shelf sparse retriever such as BM25 (Robertson et al., 2009), prioritizing low latency over the dense retriever (DR). We train for 3 epochs with $m = 4$ and $k = 5$ for WoW and 1 epoch with $m = 8$ and $k = 10$ for QReCC. $\beta$ is set to 1 for both tasks, unless stated otherwise. To ensure a thorough evaluation of ELVM's performance and its ability to generalize to unseen dialogues, we purposely avoid dataset overlap between training of R2C2-PT and ELVM. More detailed information on the training process can be found in Appendix A.

**Inference.** During inference, unlike the training phase, only a single query is generated by the query generator $p_\phi$. The default number of retrieved documents $k$ from the off-the-shelf retriever is set to 5 for WoW and 10 for QReCC. The knowledge-grounded response is generated by the response generator, $p_\theta$. An illustration of the inference is depicted in Figure 1b. During the inference, both query and response generation are conducted using beam search with a beam size of 5.

**Variants of ELVM.** To examine the impact of the proposed ELVM training, we explore another variant, *ELVM-OK*, which is trained on the response generation task with annotated *oracle knowledge* documents while keeping the query generator $p_\phi$

| | Seen | | Unseen | |
|---|---|---|---|---|
| $m$ | B4 ↑ | R2 ↑ | B4 ↑ | R2 ↑ |
| 1 | 10.8 | 12.4 | 10.5 | 12.2 |
| 2 | 11.1 | 13.0 | 11.1 | 12.8 |
| 4* | **11.9** | **13.9** | **11.8** | **13.7** |
| 8 | 11.8 | 13.7 | 11.7 | 13.6 |

Table 6: Effect of varying the number of queries, $m$, sampled during training on WoW. We see the highest performance on unseen task when $m = 4$. The super-scripted value by * is the default setting for ELVM.

| | Wow Unseen | | | QReCC | | |
|---|---|---|---|---|---|---|
| $\beta$ | B4 ↑ | R2 ↑ | R@5 ↑ | F1 ↑ | EM ↑ | R@5 ↑ |
| 0 | 9.3 | 10.6 | 65.3 | 34.5 | 5.1 | 55.8 |
| 1* | **11.8** | **13.7** | **68.3** | **36.5** | **6.9** | **65.6** |
| 5 | 10.8 | 12.4 | 66.7 | 31.3 | 3.5 | 31.9 |

Table 7: Ablation study of the posterior distillation weight $\beta$ on WoW. From the results, we can see that the trivial value of $\beta=1$ shows the best performance. The super-scripted value by * is the default setting for ELVM.

| | WoW Unseen | | | QReCC | | |
|---|---|---|---|---|---|---|
| Model | B4 ↑ | R2 ↑ | R@5 ↑ | F1 ↑ | EM ↑ | R@5 ↑ |
| ELVM | 11.8 | 13.7 | 68.3 | 36.5 | 6.2 | 59.9 |
| $\text{ELVM}_{\text{DR}}$ | 12.8 | 15.5 | 81.2 | 37.9 | 6.9 | 65.6 |

Table 8: Performance results achieved by employing the dense retriever instead of the default BM25 retriever for WoW unseen and QReCC evaluation.

| | Seen | | | Unseen | | |
|---|---|---|---|---|---|---|
| Model | B4 ↑ | R2 ↑ | R@5 ↑ | B4 ↑ | R2 ↑ | R@5 ↑ |
| R2C2-PT | 4.4 | 5.2 | 63.8 | 4.6 | 5.4 | 64.8 |
| ELVM-OK | 9.5 | 10.7 | - | 9.7 | 10.5 | - |
| ELVM | 11.9 | 13.9 | 68.4 | 11.8 | 13.7 | 68.3 |
| $\text{R2C2-PT}_{\text{Large}}$ | 8.7 | 10.2 | 62.9 | 8.9 | 10.3 | 64.4 |
| $\text{ELVM-OK}_{\text{Large}}$ | 9.6 | 10.6 | - | 10.2 | 11.2 | - |
| $\text{ELVM}_{\text{Large}}$ | 12.6 | 14.3 | 72.6 | 12.4 | 14.1 | 74.3 |

Table 9: Effect of model scaling tested on WoW. ELVM scales out to larger parameter size with increase in performance. The subscript *Large* indicates the model size of 2.7B parameters.

frozen. During inference, given a dialogue context the frozen query generator $p_\phi$ generates query and use the off-the-shelf retriever to bring relevant documents. Finally the response generator of *ELVM-OK* generates the final response.

In Section 3.2, we describe the difference between ELVM training and EM algorithm. In order to quantitatively compare these two methods, we perform EM-like training in ELVM where we apply the posterior-weighted conditional likelihood in $\mathcal{L}^{y,u}(\theta, \phi)$: $\mathcal{L}^{y,u}(\theta, \phi) = -\sum_u SG(q(u|x,y)) \log p_\theta(y|Z(u), x)$. Here, the query latents $u$ are still sampled from the prior distribution, and $SG$ means the stop-gradient that prevents the gradient propagation to the query generator. We name this variant as *ELVM-EM-Like*.

## 4.2 Automatic Evaluation

The evaluation results of WoW test seen and unseen tasks are shown in Table 2. ELVM surpasses all its variants and previous state-of-the-art (SOTA) model, achieving a new SOTA on both WoW test seen unseen tasks, across all metrics. In addition, ELVM-OK demonstrates notable improvements in BLEU and Rouge scores compared to previous models with a slight drawback in PPL. Similarly, ELVM-EM-Like also exhibits competitive performance across multiple metrics. Moreover, when comparing ELVM-from-R2C2 and ELVM, the experiment demonstrates that while R2C2-PT offers marginal benefits, the marked performance

increase is predominantly attributable to ELVM.

We further examine the performance of query generator by assessing the recall of the ground-truth documents. The results summarized in Table 3 reveal that training with our proposed algorithm leads to an increase in recall for both ELVM-$\beta$=0 and ELVM compared to R2C2-PT while the latter yields a larger performance gain. This highlights the effectiveness of distilling the posterior distribution of the response generator into the prior distribution associated with the query generator.

For QReCC, as shown in Table 4, ELVM surpasses the previous SOTA, QKConv (Cai et al., 2023), the recent previous works (Raposo et al., 2022; Kim and Kim, 2022), and the variants of ELVM including R2C2-PT and ELVM-OK in terms of all metrics. The recall performance on QReCC is included in Table C.

## 4.3 Human Evaluation

To gauge performance across multiple aspects of the quality of generated responses, human evaluation is performed on the generated responses of ELVM and KnowledGPT (Zhao et al., 2020)[1]. Similar to (Rashkin et al., 2021) and (Xu et al., 2023), we assess the response quality in two aspects: *Fluency* and *Relevance*. *Fluency* measures whether the response is understandable, self-consistent without repetition, and proficient. *Relevance* assesses the extent to which the response aligns with the dialogue context, incorporates pertinent knowledge, and maintains appropriateness. In total, 50 data

---
[1] We choose KnowledGPT because it is the best performing model among publicly available models.

| | Seen | | | Unseen | | |
|---|---|---|---|---|---|---|
| Model | PPL ↓ | B4 ↑ | R2 ↑ | PPL ↓ | B4 ↑ | R2 ↑ |
| CoLV | 39.6 | 2.9 | 7.9 | 54.3 | 2.1 | 6.3 |
| KnowledGPT | 19.2 | 7.2 | 7.9 | 22.3 | 6.0 | 6.7 |
| SPI | 17.1 | 7.7 | 8.8 | 19.1 | 7.3 | 8.5 |
| BART | 974 | 1.9 | 2.6 | 973 | 1.3 | 2.6 |
| ELVM-from-BART | 15.3 | 8.4 | 11.1 | 16.4 | 8.2 | 10.9 |
| ELVM | 14.6 | 11.9 | 13.9 | 18.0 | 11.8 | 13.7 |

Table 10: Performance comparison with BART as our initial model.

| | | | WoW Unseen | | | |
|---|---|---|---|---|---|---|
| Model | Train Ret | Test Ret | PPL ↓ | B4 ↑ | R2 ↑ | F1 ↑ |
| CoLV | - | - | 54.3 | 2.1 | 6.3 | 18.5 |
| KnowledGPT | - | - | 22.3 | 6.0 | 6.7 | 20.5 |
| SPI | - | - | 19.1 | 7.3 | 8.5 | - |
| ELVM | BM25 | BM25 | 18.0 | 11.8 | 13.7 | 27.0 |
| ELVM | BM25 | Dense | 14.8 | 12.6 | 15.2 | 29.3 |
| ELVM | Dense | Dense | 12.9 | 12.8 | 15.5 | 29.0 |
| ELVM | Dense | BM25 | 21.6 | 11.8 | 13.1 | 28.2 |

Table 11: Impact of alternating the retriever between training and testing phases on WoW unseen test. Dense indicates utilizing all-MiniLM-L6-v2 retriever as described in Section 4 and Ret is an abbreviation for retriever.

samples are randomly selected from WoW tasks where 25 samples are randomly selected from each seen and unseen task. The qualities of the responses are measured by A/B testing on the two aspects. The feedback is collected from 11 human experts. Further details and annotator instructions can be found in Table G. As shown in Table 5, ELVM significantly outperforms KnowledGPT in all aspects especially on the unseen task.

### 4.4 Ablation Studies

**Number of Queries.** The size of sampled query set $Q^s$, $m$, plays a huge role in training of ELVM, as it directly influences the bias of the marginal likelihood of the knowledge-grounded response $y$. To investigate the impact of $m$, we gradually increase $m$ from 1 to 8 and measure the performance on WoW while keeping other hyperparameters fixed. As shown in Table 6, we observe a positive correlation where increasing $m$ leads to improved performance. Notably, setting $m = 4$ yields the optimal performance for both seen and unseen tasks.

**Posterior Distillation Weight.** We examine the influence of the posterior distillation loss on ELVM training by adjusting the value of $\beta$. The experimental results presented in Table 7 reveal that the optimal performance is attained when $\beta = 1$ for both the WoW unseen and QReCC tasks, indicating the effectiveness of incorporating the posterior distillation loss. However, a substantial decline in

performance is observed when $\beta$ is increased to 5 or set to 0, particularly in the case of QReCC.

**Varying the Retriever.** We extend our experiments by incorporating the DR in two distinct ways. For WoW, we train and evaluate using all-MiniLM-L6-v2[2] DR. In contrast, for QReCC, due to its extensive knowledge pool, we leverage DR solely during evaluation by employing both the BM25 retriever and all-mpnet-base-v2[3] DR in a hybrid manner to retrieve the top-$k$ documents. Concretely, we score each document by summing up the normalized score of each retriever and re-rank to obtain top-$k$ documents. Table 8 presents a summary of the performance results of employing DR, demonstrating a successful integration of DR in different settings of ELVM.

**Scaling Up Model.** We investigate the impact of increasing the model size on performance. Starting with the publicly available 2.7B R2C2 pre-trained transformer, we follow exactly the same procedure outlined in Section 3.2 to obtain R2C2-PT$_{Large}$, ELVM-OK$_{Large}$ and ELVM$_{Large}$. Detailed hyperparameters for training 2.7B model is given in Appendix A. The results presented in Table 9 demonstrate that as the model size increases, there is an improvement in performance across all metrics, supporting the scalability of ELVM.

**BART as Base.** To ensure a fair comparison and further demonstrate the robustness of our algorithm on diverse model scales, we conduct an experiment employing BART (Lewis et al., 2019) base as an initial model and then apply our proposed algorithm on WoW dataset. We choose BART base model since the previous SOTA model (Xu et al., 2023) utilized BART base as their base model. The outcome of this training is termed ELVM-from-BART. As shown in Table 10, the initial BART model shows very poor performances. However, upon training the BART-base with our ELVM algorithm, there is a great improvement in performance across both WoW Seen and Unseen tasks. Importantly, the results surpass the performance benchmarks set by the previous SOTA model (SPI).

**Retriever Switching** To assess the robustness of our algorithm to the retriever mismatch between training and inference phases, we undertake an

---

[2]https://huggingface.co/sentence-transformers/all-MiniLM-L6-v2

[3]https://huggingface.co/sentence-transformers/all-mpnet-base-v2

| | | |
|---|---|---|
| **Dialogue Context** | I want to teach my family about investing by buying them a few shares of stock, but I really don't know much about it. | |
| | The stock market can also be called the equity market or share market. That is where you buy and sell stocks. The stock represent ownership claims on businesses. | |
| | Oh, so I could tell my kids they own part of a company. That's cool. Maybe I should get them a share in a company they know then, like Apple or XBox or Snapchat? | |
| **GT knowledge** | [...] Examples of the latter include shares of private companies which are sold to investors through equity crowdfunding platforms. [...] | |
| **GT Response** | Yes, it's like the shares of a company are sold to investors through equity crowdfunding platforms. | |
| **KnowledGPT** | **Query**: n/a | **GT knowledge selected**: False |
| | **Response**: yes, you can get shares in a company like apple or xbox. you can also get shares in company like stock market. | |
| **R2C2-PT** | **Query**: How to teach kids about investing | **GT knowledge selected**: False |
| | **Response**: The stock market can also be called the equity market or share market. That is where you buy and sell stocks. The stock represent ownership claims on businesses. | |
| **ELVM-$\beta = 0$** | **Query**: Investing tips for beginners | **GT knowledge selected**: False |
| | **Response**: That's a good idea. The stock market is a place where you can buy and sell stocks. | |
| **ELVM w/ BM25** | **Query**: Apple xbox snapchat stock share buy apple stock instagram stock investing | **GT knowledge selected**: True |
| | **Response**: That is a good idea. You can also buy shares of private companies which are sold to investors through equity crowdfunding platforms. | |
| **ELVM w/ DR** | **Query**: Best stocks for teaching kids about investing | **GT knowledge selected**: True |
| | **Response**: That is a good idea. The latter include shares of private companies which are sold to investors through equity crowdfunding platforms. | |

Table 12: Generated samples from WoW unseen task. The first row represents the dialogue context, ground truth (GT) knowledge (highlighted in olive), and GT response. Following rows contain the model generated query, response and whether the GT knowledge is included in top-5 retrieved documents. More samples can be found in Appendix H. We truncate the GT knowledge and display only relevant spans of the GT knowledge to the ground-truth response.

additional experiment, of which the results are depicted in Table 11. In this experiment, we train the WoW task utilizing either BM25 or dense retriever (utilizing all-MiniLM-L6-v2 retriever, as detailed Section 4), and subsequently evaluate using either BM25 or dense retriever. The results on the WoW unseen test indicate that utilizing a different retriever during testing doesn't necessitate model re-training, particularly given the superior performance of the dense retriever compared to BM25.

## 4.5 Qualitative Analysis

For qualitative analysis, we examine the generated samples from KnowledGPT, R2C2-PT and other variants of ELVM, whose samples are presented in Table 12. The example reveals that the queries generated by KnowledGPT, R2C2-PT and ELVM-$\beta$=0 fail to retrieve the relevant knowledge effectively while the queries generated by ELVM with BM25 and DR (all-MiniLM-L6-v2) reflects the ability to access the relevant knowledge, leading to improved responses. The results also demonstrate that the query generator trained with our algorithm effectively adapts to the chosen retriever.

## 5 Conclusion

In this paper, we propose ELVM as an efficient latent variable modeling for knowledge-intensive dialogue generation. In ELVM, the knowledge retrieval is realized by the query generation followed by document retrieval using the off-the-shelf retriever for ease of joint training of both the retriever and the response generator, without the knowledge supervision. Furthermore, the training scheme of ELVM modifies the ELBO for the marginal likelihood to effectively perform the joint training without the use of complex posterior sampling, which also eliminates the discrepancy between the training-time and test-time knowledge samplings. ELVM empirically demonstrates that it significant outperforms over previous latent variable methods with in-depth analysis on diverse dialogue datasets.

## Limitations

There are a number of limitations and possible directions to future works of ELVM proposed in this paper. First, as more large language models beyond the scale of hundreds of billion parameters are emerging, there are rooms for emergent behaviors in ELVM as the sizes of data and model grow that

has not been tested in this paper. Second, instead of the prior sampling that we use, fast posterior sampling such as short-run MCMC (Erik et al., 2019; Xu et al., 2023) is also an alternative candidate. Third, the experiments in this paper retrieve documents from confined set (e.g., Wikipedia) using light retriever such as BM25 and dense retriever. However, real-world applications often rely on the web which has much bigger pool of documents from broader open-domain. In turn, we should consider future extension of this work to utilizing larger public search engines. Last, we may consider alternative yet more effective means of query sampling during training other than a small set of random samples, as the current design. Other possible methods include imposing additional guidance for query sampling or multiple refinements of query sampling and knowledge retrieval for more diverse set.

## Ethics Statement

This work presents no direct ethical issues. However, there is a possibility that the generated responses may inadvertently contain inadequate, biased, or discriminatory content. We do not introduce new datasets, and all experiments are conducted using publicly available datasets.

## Acknowledgments

This work was supported by Institute of Information & communications Technology Planning & Evaluation (IITP) grant funded by the Korea government (MSIT)(No. 2019-0-00079, Artificial Intelligence Graduate School Program, Korea University). In addition, this work was also supported by Artificial intelligence industrial convergence cluster development project funded by the Ministry of Science and ICT(MSIT, Korea) Gwangju Metropolitan City. Finally, we would also like to thank Brain Cloud Team at Kakao Brain for their support.

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

# A  Training Details

| Parameter | Value |
|---|---|
| Training Steps | 10,000 |
| Batch Size | 128 |
| Optimizer | AdamW |
| Learning Rate (LR) | 5e-5 |
| LR Scheduler | Cosine |
| Adam $\beta_1$ | 0.9 |
| Adam $\beta_2$ | 0.999 |
| Adam $\epsilon$ | 1e-8 |
| Warmup Steps | 1000 |

Table 13: Hyperparameters used for stage-1 training. $k$ indicates number of retrieved documents.

| | WoW | QReCC |
|---|---|---|
| Train Epochs | 3 | 1 |
| Batch Size | 64 | 64 |
| Optimizer | AdamW | AdamW |
| Learning Rate (LR) | 1e-4 | 5e-4 |
| LR Scheduler | Cosine | Cosine |
| Adam $\beta_1$ | 0.9 | 0.9 |
| Adam $\beta_2$ | 0.999 | 0.999 |
| Adam $\epsilon$ | 1e-8 | 1e-8 |
| Warmup Steps | 100 | 0 |
| Min Query Length | 1 | 1 |
| Max Query Length | 32 | 32 |
| Top-k | 100 | 100 |
| Top-p | 0.95 | 0.5 |
| Temperature | 1.0 | 2.0 |

Table 14: Hyperparameters used for training ELVM and its.

Table 13 we provide hyperparameters utilized during this training process. We also report hyperparameters used for training ELVM and its variants model (ELVM-OK and ELVM-EM-Like) in Table 14. In addition, during training of QReCC, we adopt a strategy to decrease the knowledge pool size from 54M to 1M. This reduction aims to mitigate latency issues during document retrieval. Concretely, the 1M pool is constructed by initially gathering the ground-truth relevant passage for each instance in the training dataset then adding randomly sampled documents. During the evaluation stage, we utilize the original full-size knowledge pool of 54M passages.

# B  Upper Bound Performance

We believe in the value of understanding the utmost capability in an ideal scenario. To this end, we pursue an upper-bound performance analysis for both the WoW and QReCC tasks. This approach

| Model | Seen | | | | | Unseen | | | | |
|---|---|---|---|---|---|---|---|---|---|---|
| | PPL ↓ | B3 ↑ | B4 ↑ | R1 ↑ | R2 ↑ | PPL ↓ | B3 ↑ | B4 ↑ | R1 ↑ | R2 ↑ |
| ELVM | 14.6 | 14.7 | 11.9 | 29.3 | 13.9 | 18.0 | 14.5 | 11.8 | 29.2 | 13.7 |
| ELVM-w-GT-Doc | 9.7 | 22.1 | 18.5 | 40.9 | 23.9 | 10.5 | 21.5 | 17.9 | 40.7 | 23.3 |

Table 15: Performance comparison in WoW in the presence of ground-truth document during evaluation (ELVM-w-GT-Doc).

| Model | F1 ↑ | EM ↑ |
|---|---|---|
| Q.Rewriting (Raposo et al., 2022) | 18.9 | 1.0 |
| DPR-IHN (Kim and Kim, 2022) | 30.4 | 4.7 |
| QKConv (Cai et al., 2023) | 33.5 | 5.9 |
| ELVM | 36.5 | 6.2 |
| ELVM-w-GT-Doc | 55.4 | 16.2 |

Table 16: Performance comparison in QReCC in the presence of ground-truth document during evaluation (ELVM-w-GT-Doc).

| Model | Seen | | | | Unseen | | | |
|---|---|---|---|---|---|---|---|---|
| | B4 ↑ | R2 ↑ | R@5 ↑ | R@10 ↑ | B4 ↑ | R2 ↑ | R@5 ↑ | R@10 ↑ |
| $k=5$ | 11.9 | 13.9 | 68.4 | 80.3 | 11.8 | 13.7 | 68.3 | 80.3 |
| $k=5_{Bulk}$ | 9.4 | 10.9 | 62.7 | 73.9 | 8.2 | 9.5 | 47.2 | 61.9 |
| $k=10$ | 10.3 | 12.2 | 64.8 | 76.4 | 10.6 | 12.7 | 64.8 | 76.9 |
| $k=10_{Bulk}$ | 10.2 | 12.2 | 62.1 | 74.1 | 9.0 | 10.5 | 47.2 | 61.6 |

Table 18: Effect of increasing the document pool size to 1000 on the performance of ELVM. As before, $k$ indicates the number of retrieved documents used during train and inference.

involves training our response generator under the assumption that the model has access to the ground-truth document during both training and evaluation. The performance metrics of this specific setup are designated as "ELVM-w-GT-Doc" in Table 15 and Table 16. From both tables, it's evident that there still remains a margin for improvement for both tasks.

## C  Document Recall Performance

| Query from | R@1 | R@5 | R@10 |
|---|---|---|---|
| Dialogue History | 33.4 | 58.6 | 68.7 |
| Last Utterance | 25.5 | 50.8 | 62.6 |
| R2C2-PT | 17.4 | 33.0 | 39.5 |
| ELVM-$\beta$=0 | 30.6 | 55.8 | 65.7 |
| ELVM | 35.2 | 59.9 | 68.6 |
| ELVM$_{DR}$ | 37.5 | 65.6 | 75.1 |

Table 17: Query performance comparison in QReCC for different types of queries.

We report the document recall performance on QReCC, including different types of queries, as show in Table 17. Our proposed model, ELVM, surpasses both its variants and non-model generated queries such as using dialogue context as a query. The document recall performance on QReCC, encompassing various query types, is presented in Table 17. Our proposed ELVM model outperforms its variants as well as non-model generated queries, including the use of dialogue context as a query. Moreover, incorporating the dense retriever for re-ranking documents (ELVM$_{DR}$) leads to further improvements in recall.

## D  Knowledge Pool Scaling

In order to create a more realistic and challenging retrieval scenario, we expand the original setting of the WoW dataset by increasing the number of relevant documents per instance to 1000. This modification better reflects real-world information retrieval scenarios where a vast array of both relevant and irrelevant documents are typically encountered. We construct this extended dataset by randomly augmenting 1000 - $K$ documents from the WoW dataset to the document pool for each instance, where $K$ represents the number of annotated relevant documents for each instance. On average, $K \approx 20$ for each instance in WoW dataset.

Results in Table 18 show performances of ELVM models with different number of retrieved documents, $k$, and knowledge pool size, where models with increased knowledge pool has tailing keyword *Bulk* at then end of their names. Despite the increase in complexity of the retrieval task, ELVM demonstrates robust performance, with only a marginal drop in metrics compared to the original setting. Furthermore, it is important to highlight that even under these more difficult conditions with 1000 relevant documents, ELVM still outperforms SPI by a significant margin.

## E  Effect of Parameter Sharing

In our ELVM training, parameters are shared between the query and response generators. To probe this design choice, we decoupled these parameters, resulting in the ELVM-Decouple configuration, as detailed in Table 19. Upon evaluation on WoW's test seen and unseen tasks, ELVM-

| Model | Seen | | | Unseen | | |
|---|---|---|---|---|---|---|
| | PPL ↓ | B4 ↑ | R2 ↑ | PPL ↓ | B4 ↑ | R2 ↑ |
| ELVM | 14.6 | 11.9 | 13.9 | 18.0 | 11.8 | 13.7 |
| ELVM-Decouple | 16.2 | 12.6 | 14.5 | 21.6 | 12.2 | 14.0 |

Table 19: Performance comparison on the WoW dataset using the ELVM algorithm. ELVM denotes training with shared parameters between query and response generators, while ELVM-Decouple denotes a non-shared parameter approach.

| Model | Seen | | Unseen | |
|---|---|---|---|---|
| | F1 ↑ | KF1 ↑ | F1 ↑ | KF1 ↑ |
| CoLV | 20.3 | 18.2 | 18.5 | 17.5 |
| KnowledGPT | 22.0 | 23.8 | 20.5 | 22.1 |
| SPI | - | - | - | - |
| R2C2-PT | 15.4 | 20.5 | 15.7 | 22.3 |
| ELVM-OK | 23.0 | 33.0 | 22.9 | 33.0 |
| ELVM-EM-Like | 24.5 | 33.2 | 23.5 | 33.5 |
| ELVM | 27.2 | 40.8 | 27.0 | 40.3 |

Table 20: F1 and KF1 scores for ELVM and baseline models.

Decouple showed a slight edge over the standard ELVM. Nevertheless, the benefits of shared parameters—particularly regarding computational efficiency and knowledge transfer during training and inference—cannot be understated.

## F  Additional Metrics

We report the evaluation of our models on the WoW task using both F1 and knowledge F1 (KF1) metrics in Table 20. For reference, values pertaining to previous models are derived from Sun et al. (2023).

## G  Human Evaluation

For a comprehensive evaluation, we conduct the human evaluation between ELVM and KnowledGPT on seen and unseen tasks in WoW (Dinan et al., 2018). We randomly select 50 samples (25 samples per task), and each sample is evaluated by 11 different human experts. In specific, the two generated responses from each model in the same context are assigned to the annotators. For A/B testing, we give one score to the model if it's response is received an equally good or better than the other one. Figure 2 shows the annotator instructions for the two aspects.

## H  Generated Samples

Generated samples for the WoW test unseen are presented in Table 21 and Table 22. Similarly, Table 23 and Table 24 showcase the generated samples for the QReCC test.

| Dialogue Context | Hey, do you like cardigans? |
|---|---|
| | I like the modern version without no buttons and hangs open by design. |
| | Me too. My fav is the waterfall style but it's going out of fashion now I think |
| GT knowledge | [... ] Cardigan (sweater) By contrast, a pullover does not open in front but must be "pulled over" the head to be worn. [... ] |
| GT Response | But the pullover does not open in front but must be "pulled over" the head to be worn. |
| **KnowledGPT** | **Query**: n/a |
| | **GT knowledge selected**: False |
| | **Response**: i like the hand-knitted version. it's more comfortable. i like mine made of a cardigan. |
| **R2C2-PT** | **Query**: fallen fashion |
| | **GT knowledge selected**: False |
| | **Response**: I'm sorry, I didn't mean to offend you. |
| **ELVM-$\beta = 0$** | **Query**: fallen fashion cardigans |
| | **GT knowledge selected**: True |
| | **Response**: I like the one that is tied is instead considered a robe. |
| **ELVM w/ BM25** | **Query**: fallen waterfall cardigan style fallout cardigans |
| | **GT knowledge selected**: True |
| | **Response**: I like the pullover style that does not open in front but must be "pulled over" the head to be worn. |
| **ELVM w/ DR** | **Query**: fallen waterfall cardigan |
| | **GT knowledge selected**: True |
| | **Response**: I like the pullover that does not open in front but must be "pulled over" the head to be worn. |
| Dialogue Context | [...] Was she diagnosed as a child? It is much easier to diagnose children, since the symptoms begin occurring early in life |
| | Yes it started when she was 6 years old. She was 23 when I met her she was already recieving treatment |
| GT knowledge | [... ] These fellowships include pediatric neurosurgery, trauma/neurocritical care, functional and stereotactic surgery, surgical neuro-oncology, radiosurgery, neurovascular surgery, skull-base surgery, peripheral nerve and spine surgery. [... ] |
| GT Response | My first fellowship was in pediatric neurosurgery. It is great that she is doing better. |
| **KnowledGPT** | **Query**: n/a |
| | **GT knowledge selected**: False |
| | **Response**: did she cry during the periods of depression? she was crying, a negative outlook on life, and poor eye contact with others. |
| **R2C2-PT** | **Query**: types of epilepsy |
| | **GT knowledge selected**: False |
| | **Response**: I am glad to hear she is receiving treatment. Dementia is such a devastating disease. |
| **ELVM-$\beta = 0$** | **Query**: types of epilepsy |
| | **GT knowledge selected**: False |
| | **Response**: I am glad she is receiving treatment. Dementia has a significant effect on a person's caregivers. |
| **ELVM w/ BM25** | **Query**: neurosurgery epilepsy neurocognitive neurosciences |
| | **GT knowledge selected**: True |
| | **Response**: I am glad to hear that she is receiving treatment. Neurosurgery is a branch of medicine that specializes in the treatment of epilepsy. |
| **ELVM w/ DR** | **Query**: neurosurgery treatment for epilepsy |
| | **GT knowledge selected**: True |
| | **Response**: Did she undergo a fellowship? Many neurosurgeons pursue additional training in the form of a fellowship, after residency or in some cases, as a senior resident. |

Table 21: Generated samples from WoW test unseen. We truncate the ground-truth knowledge and display only relevant spans of the ground-truth knowledge to the ground-truth response.

| | |
|---|---|
| **Dialogue Context** | Do you know anything about the band called Insane Clown Posse? |
| | Insane Clown Posse (ICP) is an American hip hop duo composed of Violent J ,and Shaggy 2 Dope. |
| | [...] |
| | That's quite a lot of albums, what else do you know about ICP? |
| **GT knowledge** | [... ] The duo founded the independent record label Psychopathic Records with Alex Abbiss as manager, and produced and starred in the feature films "Big Money Hustlas" and "Big Money Rustlas". [... ] |
| **GT Response** | The duo founded the independent record label Psychopathic Records with Alex Abbiss as manager, and produced and starred in the feature films "Big Money Hustlas |
| **KnowledGPT** | **Response: Query**: n/a |
| | **GT knowledge selected**: False |
| | **Response**: the duo has earned two platinum and five gold albums. they have also earned two platinum and five gold albums |
| **R2C2-PT** | **Query**: ICP wiki |
| | **GT knowledge selected**: False |
| | **Response**: The duo has earned two platinum ,with 5 golden album |
| **ELVM-$\beta = 0$** | **Query**: ICP facts |
| | **GT knowledge selected**: False |
| | **Response**: The group has been around for a long time . |
| **ELVM w/ BM25** | **Query**: Insane clown posse ICP facts |
| | **GT knowledge selected**: True |
| | **Response**: The duo founded the independent record label Psychopathic Records with Alex Abbiss as manager, and produced and starred in the feature films "Big Money Hustlas" |
| **ELVM w/ DR** | **Query**: Insane clown posse facts |
| | **GT knowledge selected**: True |
| | **Response**: The duo founded the independent record label Psychopathic Records with Alex Abbiss as manager, and produced and starred in the feature films "Big Money Hustlas" |
| **Dialogue Context** | [...] Athletes normally eat meat in large quantities. |
| | Exactly. I have also added much meat eating as part of my special diet. They are rich in protein. Especially, red meat |
| **GT knowledge** | [... ] In nutritional science, on the other hand, red meat is defined as any meat that has more myoglobin than white meat, white meat being defined as non-dark meat from chicken (excluding leg or thigh), or fish. [... ] |
| **GT Response** | Red meat is divined as any meat that has myoglobin than white |
| **KnowledGPT** | **Query**: n/a |
| | **GT knowledge selected**: False |
| | **Response**: red meat is classified as white meat. i think it's interesting that only flesh from mammals or fowl is classified as red. |
| **R2C2-PT** | **Query**: protein powder |
| | **GT knowledge selected**: False |
| | **Response**: Athletes normally eat meat in large quantities. |
| **ELVM-$\beta = 0$** | **Query**: red meat protein |
| | **GT knowledge selected**: True |
| | **Response**: In nutritional science, red meat is defined as any meat that has more myoglobin than white meat. |
| **ELVM w/ BM25** | **Query**: red meat protein |
| | **GT knowledge selected**: True |
| | **Response**: In nutritional science, red meat is defined as any meat that has more myoglobin than white meat. |
| **ELVM w/ DR** | **Query**: red meat protein |
| | **GT knowledge selected**: True |
| | **Response**: Red meat is defined as any meat that has more myoglobin than white meat. |

Table 22: Generated samples from WoW test unseen. We truncate the ground-truth knowledge and display only relevant spans of the ground-truth knowledge to the ground-truth response.

| Dialogue Context | When did Mark Spitz' swimming career end? |
| | Following the Munich Olympics |
| | [...] |
| | What else did Mark Spitz do besides Speedo swimsuit ads after his swimming career? |
| GT knowledge | [...] However, as the memory of his feats receded, so did his endorsement and promotional deals. As his endorsements faded, he started a successful real-estate company in Beverly Hills. He bought a Ferrari and says he made [...] |
| GT Response | Mark Spitz started a successful real-estate company in Beverly Hills |
| **R2C2-PT** | **Query**: What did mark spitz do after his olympic career? |
| | **GT knowledge selected**: False |
| | **Response**: In 1999 Spitz ranked No. 33 on ESPN SportsCentury 50 Greatest Athletes, the only aquatic athlete to make the list. |
| **ELVM-$\beta = 0$** | **Query**: Mark Spitz Speedo Swimsuit ads after speedo swimming career ends mark spitz m Munich olympics 2012 |
| | **GT knowledge selected**: False |
| | **Response**: Spitz did Schick razor commercials and Speedo swimsuit ads |
| **ELVM w/ BM25** | **Query**: Spitz endorsements after swimming career ends |
| | **GT knowledge selected**: True |
| | **Response**: Mark Spitz started a successful real-estate company in Beverly Hills. |
| Dialogue Context | What was the New Scheme? |
| | The 'new scheme' and the 'new dedication' respectively, initially seen as a [...] |
| | [...] |
| | Did Ahmadi followers of the New Scheme act as missionaries? |
| | As part of this Mahmood Ahmad appointed 5 men to survey the Punjab [...] |
| | Did Mirza Basheer-ud-Din Mahmood Ahmad use missionaries to try to spread the New Scheme to foreign countries? |
| GT knowledge | [...] In time the scheme produced a vast amount of literature in defence of Islam in general and the Ahmadiyya beliefs in particular. The funds were also spent on the training and dispatching of Ahmadiyya missionaries outside the Indian sub-continent and their sustenance. As part of this Mahmood Ahmad appointed [...] |
| GT Response | The funds were also spent on the training and dispatching of Ahmadiyya missionaries outside the Indian sub-continent and their sustenance. |
| **R2C2-PT** | **Query**: Mirza Basher-ud Din Mahmood Ahmad missionaries |
| | **GT knowledge selected**: False |
| | **Response**: Mirza Basheer-ud-Din Mahmood Ahmad used missionaries to try to spread the New Scheme to foreign countries. |
| **ELVM-$\beta = 0$** | **Query**: New Scheme Mahmud Ahmad missionaries to spread New Scheme in foreign countries |
| | **GT knowledge selected**: True |
| | **Response**: Yes, I have. I have experienced heartache, also known as loss of a loved one. |
| **ELVM w/ BM25** | **Query**: New scheme missionaries used to spread Ahmadiyyah teachings in foreign countries |
| | **GT knowledge selected**: True |
| | **Response**: The funds were also spent on the training and dispatching of Ahmadiyya missionaries outside the Indian sub-continent and their sustenance. |

Table 23: Generated samples from QReCC test. We truncate the ground-truth knowledge and display only relevant spans of the ground-truth knowledge to the ground-truth response.

| Dialogue Context | How did Stephen Sondheim collaborate with James Lapine? |
|---|---|
| | Stephen Sondheim found a new collaborator in [...] |
| | What did Stephen Sondheim and James Lapine do together in the 1980s? |
| | Their first collaboration was Sunday in the Park with George (1984). |
| | Was Stephen Sondheim and James Lapine's first collaboration a success? |
| | Stephen Sondheim and James Lapine won the 1985 Pulitzer Prize [...] |
| | Did Stephen Sondheim and James Lapine collaborate in other projects, besides Sunday in the Park with George? |
| | [...] last work together was the rhapsodic Passion (1994). |
| | Was Stephen Sondheim and James Lapine's last work a success? |
| GT knowledge | Sondheim and Lapine's last work together was the rhapsodic Passion (1994), adapted from Ettore Scola's Italian film Passione D'Amore. With a run of 280 performances, Passion was the shortest-running show to win a Tony Award for Best Musical.[46] Later work Assassins opened [...] |
| GT Response | With a run of 280 performances, Passion was the shortest-running show to win a Tony Award for Best Musical. |
| **R2C2-PT** | **Query**: Was the play "rhapsodic passion" a success or a failure? |
| | **GT knowledge selected**: True |
| | **Response**: Sondheim and Lapine's last work together was the rhapsodic Passion (1994). |
| **ELVM-$\beta = 0$** | **Query**: Stephen Sonds and Lapine Rhapsodic passion1994 Stephen sondheim, James lapine plays 12 dreams off-broadway p Pulitzer prize for play Sunday in Park. |
| | **GT knowledge selected**: False |
| | **Response**: Stephen Sondheim and James Lapine's last work together was the rhapsodic Passion (1994). |
| **ELVM w/ BM25** | **Query**: Sondheim last work was the Rhapsodic passion (1994) Pulitzer prize won for Drama. Lapine wrote the play Twelve Dreams. |
| | **GT knowledge selected**: True |
| | **Response**: With a run of 280 performances, Passion was the shortest-running show to win a Tony Award for Best Musical. |
| Dialogue Context | Did Frank Hague commit any questionable acts? |
| | Yes, he took kickbacks in cash, and relied on two ordinances of dubious [...] |
| | Did Frank Hague do anything illegal? |
| | He passed an unconstitutional ordinance that gave himself as commissioner the power to [...] |
| | Did anyone attempt to prosecute Frank Hague? |
| GT knowledge | [...] Hague was able to stay in power despite a nearly constant effort to turn him out of office from 1921 onward. [38] He was also able to avoid prosecution despite numerous federal and state investigations in part due to the fact he took most of his kickbacks in cash. [...] |
| GT Response | Yes, but he was also able to avoid prosecution despite numerous federal and state investigations in part due to the fact he took most of his kickbacks in cash. |
| **R2C2-PT** | **Query**: Frank Hague prosecution |
| | **GT knowledge selected**: False |
| | **Response**: No, no one tried to prosecute Frank Hague. |
| **ELVM-$\beta = 0$** | **Query**: Frank Hague ordinance ordinance frank Hague charges Frank Hague Hague for flouting constitutionality ordinances |
| | **GT knowledge selected**: False |
| | **Response**: Hague was found guilty of contempt of court for ignoring the subpoena. He was fined $100 and stripped of his duties as Deputy Sheriff. |
| **ELVM w/ BM25** | **Query**: Hague prosecution attempt failed after Hague passed unconstitutional ordinance. no Hague was prosecuted. |
| | **GT knowledge selected**: True |
| | **Response**: Hague was able to stay in power despite a nearly constant effort to turn him out of office from 1921 onward. He was also able to avoid prosecution despite numerous federal and state investigations in part due to the fact he took most of his kickbacks in cash. |

Table 24: Generated samples from QReCC test. We truncate the ground-truth knowledge and display only relevant spans of the ground-truth knowledge to the ground-truth response.

# Human Evaluation Annotation Tool

## Components of each example

Each example may include one of more of the following

- History: topic related to the conversation and the dialogue history. Examples may not contain 'history'
- Input text: Utterance of the opponent
- Ground truth response of the example, given as a reference. The annotator may use the reference in decision.
- Two esponses from different models.

## Annotation Standards

The annotator marks which model of the given two is better on three different categories: Fluency and Relevance. The deifintion of each category is

- Fluency measures the correctness, completeness, and semantic consistency of the response.
- Relevance measures the degree of relevance of the knowledge and response to the history.

---

*History:*
Singing
I love to sing. Some of my favorite genres to sing are blues, pop, and rock. Do you like to sing?
Yes very much so. I love older artists like Ray Charles and Billie Holiday.
Oh yeah I love Billie Holiday. She had no formal music training but her vocal delivery made up for it.

*Input Text:*
I've been getting into the strokes as of late. Do you like them?

*Reference:*
I used to listen to The Strokes a lot back in high school. Their debut album was added to Rolling Stones 500 greatest albums of all time at number 199.

---

*Model A Response:*
Yes I do like them. I like all kinds of music, but I'm not a big fan of the Strokes.

*Model B Response:*
I do like them. I also like to wear my Strokes shirt during May because it is national stroke awareness month.

# Annotation

Fluency

- ● Both responses are fluent.
- ○ Model A response is more fluent.
- ○ Model B response is more fluent.
- ○ Neither response is fluent.

Relevance

- ● Both knowledges and responses are relevant.
- ○ Model A response is more relevant.
- ○ Model B response is more relevant.
- ○ Neither is relevant.

Previous    Next

Made with Streamlit

Figure 2: Screenshot of the annotation tool.