# OpenReview forum: "Efficient Latent Variable Modeling for Knowledge-Grounded Dialogue Generation"
_EMNLP/2023/Conference — EMNLP 2023 Findings_

### Official Review · Reviewer_UuJj · 2023-07-24

**Soundness:** 2

**Excitement:**

2: Mediocre: This paper makes marginal contributions (vs non-contemporaneous work), so I would rather not see it in the conference.

**Paper Topic And Main Contributions:**

This paper focuses on the problem setting of knowledge-grounded dialogue generation. Prior works with finetuning knowledge retrievers have constraints on the use of complex and large-scale retrievers, or train-infer discrepancy when involving posterior probability. To alleviate these limitations, this paper proposes a latent variable modeling (namely ELVM) with a query generator, an off-the-shelf knowledge retriever, and a response generator, which treats queries instead of retrieved knowledge as a latent variable and applies a modified ELBO. The proposed method achieves SOTA on two datasets.

**Questions For The Authors:**

Question A:

Why the Equation 3 (the conditional likelihood of response) not consider the term of $P(Z(u) | u)$?

Question B:

On the QReCC dataset, the response generator takes **TEN** knowledge to predict the response. However, the knowledge retrieval Recall@10 of dialogue history surpasses ELVM in Table 13. What is the effect of training the response generator directly with the dialogue history as the query?

**Reasons To Accept:**

1. The motivation of this paper is intuitive and easy to understand.
2. The proposed method is clearly clarified and makes a step in modifying the ELBO.
3. The automatic and human evaluation shows the proposed method achieves state-of-the-art results on two different datasets.

**Reasons To Reject:**

1. The contribution is unclear.

For example, in Line 143, this paper states the first contribution is a novel latent variable modeling method that trains query and response generators simultaneously via joint training. However, this contribution is ambiguous, as previous research has already proposed queries as intermedia results and joint training of query and response generators, as stated in Line 252.

2. The effectiveness of the proposed method without a pretraining process is unclear.

ELVM is initial from R2C2-PT, which is post-pretrained in many other knowledge-grounded conversation datasets. However, the baseline models compared in this paper are trained from vanilla pretrained models. This paper does not validate the method's effectiveness without post-pretraining, or compare baseline methods on R2C2-PT.

**Reproducibility:**

4: Could mostly reproduce the results, but there may be some variation because of sample variance or minor variations in their interpretation of the protocol or method.

**Reviewer Confidence:**

4: Quite sure. I tried to check the important points carefully. It's unlikely, though conceivable, that I missed something that should affect my ratings.

---

> ### Author Rebuttal · Authors · 2023-08-29
>
> We deeply appreciate the time and effort you have dedicated to reviewing our paper. Your insightful feedback has been instrumental in refining our work, and we are grateful for the constructive suggestions and the interest you've shown. We will incorporate your comments and feedback into our revised version. We have taken note of your concerns and questions and will address them in the sections below.
>
> **The contribution is unclear:**
>
> We acknowledge the potential confusion arising from the phrasing in our paper. In light of your feedback, we will have removed the term "novel" from line 143 in our revised version, and restate our first contribution as: "An efficient latent variable modeling is proposed in joint training of query generator and dialogue response generator without the knowledge supervision for knowledge-intensive dialogue generation”.  We hope these adjustments better convey our intentions and address your concerns.
>
> **Use of R2C2 in ELVM's Training:**
>
> - **Addressing the Central Concern:**
>
>     We deeply appreciate the thorough feedback regarding the possible attribution of ELVM's performance improvements to the utilization of R2C2-PT.
>
>     To rigorously address the aforementioned concern, we have undertaken an additional experiment, deliberately omitting the post-pretraining stage. Our focus was to empirically validate whether the gains in performance arise primarily from the utilization of R2C2-PT or if they can be ascribed to the efficacy of our proposed algorithm, ELVM.
>
>
> - **Experiment Design:**
>
>     We start with a R2C2 transformer model as our base rather than R2C2-PT and then train with ELVM algorithm. The inference settings remained unaltered from Section 4.1 (line 422).
>
>     To avoid ambiguity, we name this resulting model as “ELVM-from-R2C2” and rename the model trained from R2C2-PT as "ELVM-from-R2C2-PT", which corresponds to the "ELVM" terminology utilized in our original submission.
>
> - **Results:**
>
>     Our results illustrate that while the standalone R2C2 model showed poor performances, our proposed ELVM leads to great improvements (ELVM-from-R2C2). Note that this improved performance is better than that of the previous SOTA model (SPI) and is comparable to the result obtained by ELVM-from-R2C2-PT.
>
>     | Model             | WoW Seen |      |      |      |      | WoW Unseen |      |      |      |      |
>     |-------------------|:--------:|:----:|:----:|:----:|:----:|:----------:|:----:|:----:|:----:|:----:|
>     |                   |   PPL ↓  | B3 ↑ | B4 ↑ | R1 ↑ | R2 ↑ |    PPL ↓   | B3 ↑ | B4 ↑ | R1 ↑ | R2 ↑ |
>     | CoLV              |   39.6   |  -   |  2.9 | 20.6 |  7.9 |    54.3    |  -   |  2.1 | 19.7 |  6.3 |
>     | KnowledGPT        |   19.2   |  9.5 |  7.2 | 22.0 |  7.9 |    22.3    |  8.3 |  6.0 | 20.5 |  6.7 |
>     | SPI               |   17.1   | 10.2 |  7.7 | 22.7 |  8.8 |    19.1    |  9.6 |  7.3 | 22.0 |  8.5 |
>     | R2C2-PT           |   24.7   |  6.0 |  4.4 | 17.3 |  5.2 |    30.4    |  6.2 |  4.6 | 17.7 |  5.4 |
>     | ELVM-from-R2C2-PT |   14.6   | 14.7 | 11.9 | 29.3 | 13.9 |    18.0    | 14.5 | 11.8 | 29.2 | 13.7 |
>     | R2C2              |   40.7   |  1.7 |  0.9 | 12.3 |  1.9 |    46.6    |  1.6 |  0.9 | 11.9 |  5.4 |
>     | ELVM-from-R2C2    |   15.8   | 14.5 | 11.8 | 28.8 | 13.3 |    18.9    | 14.3 | 11.5 | 28.8 | 13.3 |
>
> - **Conclusions:**
>
>     From the supplementary experiment, it is discernible that while R2C2-PT leads some advantages, the significant performance enhancements can be largely credited to our proposed algorithm, ELVM. This underscores ELVM's capability to drive performance improvements.
>
> **BART as Base:**
>
> We acknowledge the valid point raised about the potential unfair advantage due to the discrepancy between our initial R2C2 model (400M parameters) and the BART-base model (140M parameters) used for the previous models including the previous SOTA model (SPI) on the WoW task.
>
> - **Addressing the Model Size Discrepancy:**
>
>     To ensure a fair comparison and further demonstrate the robustness of our algorithm on diverse model scales, we conducted an experiment employing the BART-base as an initial model and then applied our proposed algorithm, ELVM, on the WoW dataset. The outcome of this training is termed “ELVM-from-BART.” Please note that due to the limted period and resource, we conduct this experiment on WoW dataset.
>
> - **Evaluation Results:**
>
>     The initial BART model shows very poor performances. However, upon training the BART-base with our ELVM algorithm, there is a great improvement in performance across both WoW Seen and Unseen tasks. Importantly, the results surpass the performance benchmarks set by the previous SOTA model (SPI). This endorses the robustness and scalability of our algorithm, confirming its efficacy irrespective of the model size, from BART-base's scale (140M) to a billion-scale parameter size (as elaborated in Section 4.4, line 539). We believe this addresses the primary concern regarding the potential advantage due to model size and underscores the genuine contributions of our research.
>
>     | Model             | WoW Seen |      |      |      |      | WoW Unseen |      |      |      |      |
>     |-------------------|:--------:|:----:|:----:|:----:|:----:|:----------:|:----:|:----:|:----:|:----:|
>     |                   |   PPL ↓  | B3 ↑ | B4 ↑ | R1 ↑ | R2 ↑ |    PPL ↓   | B3 ↑ | B4 ↑ | R1 ↑ | R2 ↑ |
>     | CoLV              |   39.6   |  -   |  2.9 | 20.6 |  7.9 |    54.3    |  -   |  2.1 | 19.7 |  6.3 |
>     | KnowledGPT        |   19.2   |  9.5 |  7.2 | 22.0 |  7.9 |    22.3    |  8.3 |  6.0 | 20.5 |  6.7 |
>     | SPI               |   17.1   | 10.2 |  7.7 | 22.7 |  8.8 |    19.1    |  9.6 |  7.3 | 22.0 |  8.5 |
>     | R2C2-PT           |   24.7   |  6.0 |  4.4 | 17.3 |  5.2 |    30.4    |  6.2 |  4.6 | 17.7 |  5.4 |
>     | ELVM-from-R2C2-PT |   14.6   | 14.7 | 11.9 | 29.3 | 13.9 |    18.0    | 14.5 | 11.8 | 29.2 | 13.7 |
>     | R2C2              |   40.7   |  1.7 |  0.9 | 12.3 |  1.9 |    46.6    |  1.6 |  0.9 | 11.9 |  5.4 |
>     | ELVM-from-R2C2    |   15.8   | 14.5 | 11.8 | 28.8 | 13.3 |    18.9    | 14.3 | 11.5 | 28.8 | 13.3 |
>     | BART              |   974.0  |  1.9 |  1.3 | 11.2 |  2.6 |    973.0   |  1.8 |  1.3 | 10.9 |  2.6 |
>     | ELVM-from-BART    |   15.3   | 10.6 |  8.4 | 25.8 | 11.1 |    16.4    | 10.4 |  8.2 | 25.7 | 10.9 |
>
> ---
>
> **Question A. $P(Z(u)|u)$:**
>
> We appreciate your observation regarding Equation 3. Indeed, we did not incorporate the term $P(Z(u)|u)$ because it represents a deterministic mapping, analogous to a delta function. As we employ a static retriever such as BM25, rather than joint training of a document retriever, we can safely make the assumption that $P(Z(u)|u)$ equals to one.
>
> **Question B. Dialogue History as Query:**
>
> As an additional experiment, we directly train the dialogue response generator by treating the dialogue history itself as a query. During evaluation, this dialogue history is also used as the query to retrieve the top-\{ 1, 5, 10 \} most relevant documents ($k$) via the BM25 retriever. We designate this model "ELVM-Context-as-Query" and compare its performance against ELVM.
>
> In the following table, we report three values for each metric corresponding to using top-\{ 1, 5, 10 \} documents from the retriever. We can observe a significant gap between ELVM and ELVM-Context-as-Query. In other words, although using the dialogue context as a query achieves marginally better document recall than using ELVM’s generated query at $k=10$ as in Table 13, the final performance of ELVM is better than that of ELVM-Context-as-Query.
>
> | Model                 | F1↑   |      |      | EM↑  |     |      |
> |-----------------------|------|------|------|-----|-----|------|
> |                       | $k$=1  | $k$=5  | $k$=10 | $k$=1 | $k$=5 | $k$=10 |
> | QKConv                | -    | -    | 33.5 | -   | -   | 5.9  |
> | ELVM-Context-as-Query | 27.5 | 33.1 | 35.6 | 2.6 | 4.7 | 4.8  |
> | ELVM                  | 29.1 | 35.0 | 36.5 | 3.1 | 5.4 | 6.2  |
>
> In addition, we try to identify the reason of why the dialogue context can be a strong query on the QReCC task, especially when the number of retrieved documents is large. We observe that this is due to the substantial alignment between the dialogue context and the ground-truth knowledge especially in the QReCC test dataset, in contrast to the WoW test dataset. Our obtained F1 scores between the dialogue context and ground-truth knowledge support this: QReCC shows a score of 21.1, which is significantly higher than those of the WoW test seen (10.7) and unseen (11.3) tasks.

---

### Official Review · Reviewer_ivGY · 2023-08-05

**Soundness:** 3

**Excitement:**

3: Ambivalent: It has merits (e.g., it reports state-of-the-art results, the idea is nice), but there are key weaknesses (e.g., it describes incremental work), and it can significantly benefit from another round of revision. However, I won't object to accepting it if my co-reviewers champion it.

**Missing References:**

KGPT: Knowledge-Grounded Pre-Training for Data-to-Text Generation - Chen et al.

Knowledge-grounded dialogue generation with a unified knowledge representation - Li et al.

Low-resource knowledge-grounded dialogue generation - Zhao et al.

**Paper Topic And Main Contributions:**

This paper introduces an approach to knowledge-grounded dialogue generation through the development of an Efficient Latent Variable Modeling (ELVM). The model incorporates a query generator, which is then followed by an off-the-shelf retriever that is fixed during the training process. The key innovation of the ELVM is its ability to conduct efficient and robust training without the need for annotated knowledge or explicit training of knowledge retrieval. Experimental results demonstrate the effectiveness of this approach.

**Questions For The Authors:**

1. I notice that the evaluation does not include the reporting of the Knowledge F1 (KF1) metric. Could you please explain the reason behind this decision? Also, the human evaluation did not include the R2C2-PT baseline. Can you provide some clarification on this matter?

2. I notice that your method suggests fixing the knowledge retriever during training. However, it seems that if we switch to a new knowledge retriever, the model will need to be re-trained. Can you please explain how this approach contributes to the overall efficiency of the model?

**Reasons To Accept:**

1. The proposed Latent Variable Model (ELVM) introduces a significant innovation in the field of knowledge-grounded dialogue generation. Unlike traditional models, the ELVM does not require annotated knowledge or explicit training of knowledge retrieval. This feature simplifies the training process and makes the model more adaptable to various applications.

2. The training objective eliminates the need for extra modeling of a surrogate posterior distribution or online posterior inference. This simplifies the training process and improves its effectiveness.

**Reasons To Reject:**

1. While the paper introduces a novel approach in the form of the ELVM, the actual improvement over existing methods appears to be marginal.

2. The paper falls short in thoroughly analyzing the knowledge grounding performance. Key metrics such as Knowledge F1 are not evaluated.

3. It seems that the paper's human evaluation part is lacking as it does not cover all pertinent baselines for comparison.

**Reproducibility:**

3: Could reproduce the results with some difficulty. The settings of parameters are underspecified or subjectively determined; the training/evaluation data are not widely available.

**Reviewer Confidence:**

2: Willing to defend my evaluation, but it is fairly likely that I missed some details, didn't understand some central points, or can't be sure about the novelty of the work.

---

> ### Author Rebuttal · Authors · 2023-08-29
>
> We deeply appreciate the time and effort you have dedicated to reviewing our paper. Your insightful feedback has been instrumental in refining our work, and we are grateful for the constructive suggestions and the interest you've shown. We will incorporate your comments and feedback into our revised version. We have taken note of your concerns and questions and will address them in the sections below.
>
> **On Performance Improvement:**
>
> We respectfully differ with the view that our improvement is marginal. Analyzing the metrics reveals a clear trend of significant improvement. For the WoW unseen task, ELVM attains 14.5 (Bleu-3) and 29.2 (Rouge-1). This contrasts with SPI's 9.6 and 22.0 and KnowledGPT's 8.3 and 6.0 for the same metrics. Clearly, the gap between ELVM and SPI is larger than the difference between SPI and KnowledGPT.
>
> In the QReCC evaluation, our model registers 36.5 on the F1 metric, notably outperforming previous models like QKConv (33.5) and DPR-IHN (30.4). Moreoever, augmenting our model with a dense retriever during inference, akin to the methodology of QKConv and DPR-IHN, enhances our F1 score to 37.9.
>
> **On Human Evaluation:**
>
> We intended to undertake human evaluations compared to robust models like SPI. However, the unavailability of SPI's checkpoint led us to opt for KnowledGPT, a commendable performer amongst accessible models. Given the considerable expenditure linked with human evaluations, we could not include R2C2-PT in our human evaluation. Another reason for the exclusion of R2C2-PT is in the findings of Table 1, where R2C2-PT underperforms compared to KnowledGPT.
>
> ----
>
> **Question 1. Metrics Reporting:**
>
> Thank you for highlighting the importance of comprehensive metric representation in our study. We concur with your view that including metrics like Knowledge-F1 (KF1) can advance thoroughness and comprehensiveness of our analysis. In response to the feedback, we report both F1 and KF1 metrics for our models and the values of previous models are drawn from [1]. As similar to other metrics, ELVM shows better performances than the previous methods in terms of both F1 and KF1. We will ensure a comprehensive metric inclusion in the revised version.
>
> | Model        | WoW Seen |      | WoW Unseen |      |
> |--------------|:--------:|------|------------|------|
> |              |    F1↑   |  KF1 |     F1↑    |  KF1↑ |
> | CoLV         |   20.3   | 18.2 |    18.5    | 17.5 |
> | KnowledGPT   |   22.0   | 23.8 |    20.5    | 22.1 |
> | SPI          |     -    |   -  |      -     |   -  |
> | R2C2-PT      |   15.4   | 20.5 |    15.7    | 22.3 |
> | ELVM-OK      |   23.0   | 33.0 |    22.9    | 33.0 |
> | ELVM-EM-Like |   24.5   | 33.2 |    23.5    | 33.5 |
> | ELVM         |   27.2   | 40.8 |    27.0    | 40.3 |
>
> **Question 2. Retriever Switching:**
>
> To assess the robustness of our algorithm to retriever mismatch between training and inference phases, we undertake an additional experiment, of which the results are depicted in the table below. In this experiment, we train the WoW task utilizing eitherBM25 or dense retriever (utilizing the all-MiniLM-L6-v2 encoder, as detailed in line 528), and subsequently evaluate using eitherBM25 or dense retriever. The results on the WoW unseen test indicate that the use of a different retriever at the test time does not require re-training of the model, considering just the better performance by dense retriever than BM25 in general.
>
> | Model      | Train Retriever | Inference Retriever | PPL ↓ | B3 ↑ | B4 ↑ | R1 ↑ | R2 ↑ | F1↑   | KF1↑  |
> |------------|-----------------|---------------------|-------|------|------|------|------|------|------|
> | CoLV       | -               | -                   | 54.3  | -    | 2.1  | 19.7 | 6.3  | 18.5 | 17.5 |
> | KnowledGPT | -               | -                   | 22.3  | 8.3  | 6.0  | 20.5 | 6.7  | 20.5 | 22.1 |
> | SPI        | -               | -                   | 19.1  | 9.6  | 7.3  | 22.0 | 8.5  | -    | -    |
> | ELVM       | BM25            | BM25                | 18.0  | 14.5 | 11.8 | 29.2 | 13.7 | 27.0 | 40.3 |
> | ELVM       | BM25            | Dense               | 14.8  | 15.5 | 12.6 | 31.2 | 15.2 | 29.3 | 45.4 |
> | ELVM       | Dense           | Dense               | 12.9  | 15.2 | 12.8 | 31.1 | 15.5 | 29.0 | 45.6 |
> | ELVM       | Dense           | BM25                | 21.6  | 14.5 | 11.8 | 28.4 | 13.1 | 28.2 | 44.4 |
>
> [1] Generative Knowledge Selection for Knowledge-Grounded Dialogues. Sun et al.

---

### Official Review · Reviewer_unbb · 2023-08-08

**Soundness:** 3

**Excitement:**

3: Ambivalent: It has merits (e.g., it reports state-of-the-art results, the idea is nice), but there are key weaknesses (e.g., it describes incremental work), and it can significantly benefit from another round of revision. However, I won't object to accepting it if my co-reviewers champion it.

**Missing References:**

End-to-End Training of Multi-Document Reader and Retriever for Open-Domain Question Answering. In NeurIPS 2021.

**Paper Topic And Main Contributions:**

This paper proposes an efficient latent variable modeling for knowledge-grounded dialogue generation where the query generator for knowledge retrieval from the off-the-shelf retriever and the dialogue response generator grounded on the retrieved knowledge are simultaneously trained without the knowledge supervision. The proposed ELVM demonstrates its effectiveness in performing unsupervised joint training and significant performance improvements over previous methods.

**Reasons To Accept:**

1. The proposed algorithm achieves state-of-the-art performances without the use of the annotated knowledge, which ease the problem of expensive human annotation of knowledge information.
2. This paper conducts several experiments on benchmark datasets and analyzes the experimental results comprehensively.


**Reasons To Reject:**

1. In line 391, you fine-tune R2C2 pre-trained transformer model on multiple datasets to acquire the essential skill sets for knowledge-grounded dialogue such as query generation and knowledge-grounded response generation. It would be better if you could supplement what the datasets is in detail to judge if it may result in unfair experimental settings between baselines and the proposed model.
2. In the baseline of KnowledGPT, the authors employ BERT and GPT-2 for knowledge selection and dialogue generation, respectively, which is different from the model scale/pretraining corpus size employed in this paper. So I have concerns regarding whether the performance improvements in this work are attributed to the utilization of a more proficient pre-training language model.
3. Utilizing variational techniques to enhance knowledge-intensive text generation tasks is not a novel approach. Besides, it is suggested that the authors provide more illustration on the distinctions between proposed method and other similar approaches in this field (e.g. [1])

[1] End-to-End Training of Multi-Document Reader and Retriever for Open-Domain Question Answering. In NeurIPS 2021.

**Reproducibility:**

3: Could reproduce the results with some difficulty. The settings of parameters are underspecified or subjectively determined; the training/evaluation data are not widely available.

**Reviewer Confidence:**

4: Quite sure. I tried to check the important points carefully. It's unlikely, though conceivable, that I missed something that should affect my ratings.

---

> ### Author Rebuttal · Authors · 2023-08-29
>
> We deeply appreciate the time and effort you have dedicated to reviewing our paper. Your insightful feedback has been instrumental in refining our work, and we are grateful for the constructive suggestions and the interest you've shown. We will incorporate your comments and feedback into our revised version. We have taken note of your concerns and questions and will address them in the sections below.
>
> **Datasets for R2C2 Training:**
>
> We've detailed the datasets harnessed for training the R2C2 model in Table 10, Appendix A, aiming to instill crucial competencies for knowledge-grounded dialogue. Recognizing the significance of this information, we will move the table to the main section in the revised version. Respective descriptions pertaining to each dataset will be incorporated alongside.
>
> **Utilization of R2C2 in ELVM's Training:**
>
> - **Addressing the Central Concern:**
>
>     We deeply appreciate the thorough feedback regarding the possible attribution of ELVM's performance improvements to the utilization of R2C2-PT.
>
>     To rigorously address the aforementioned concern, we have undertaken an additional experiment, deliberately omitting the post-pretraining stage. Our focus was to empirically validate whether the gains in performance arise primarily from the utilization of R2C2-PT or if they can be ascribed to the efficacy of our proposed algorithm, ELVM.
>
>
> - **Experiment Design:**
>
>     We start with a R2C2 transformer model as our base rather than R2C2-PT and then train with ELVM algorithm. The inference settings remained unaltered from Section 4.1 (line 422).
>
>     To avoid ambiguity, we name this resulting model as “ELVM-from-R2C2” and rename the model trained from R2C2-PT as "ELVM-from-R2C2-PT", which corresponds to the "ELVM" terminology utilized in our original submission.
>
> - **Results:**
>
>     Our results illustrate that while the standalone R2C2 model showed poor performances, our proposed ELVM leads to great improvements (ELVM-from-R2C2). Note that this improved performance is better than that of the previous SOTA model (SPI) and is comparable to the result obtained by ELVM-from-R2C2-PT.
>
>     | Model             | WoW Seen |      |      |      |      | WoW Unseen |      |      |      |      |
>     |-------------------|:--------:|:----:|:----:|:----:|:----:|:----------:|:----:|:----:|:----:|:----:|
>     |                   |   PPL ↓  | B3 ↑ | B4 ↑ | R1 ↑ | R2 ↑ |    PPL ↓   | B3 ↑ | B4 ↑ | R1 ↑ | R2 ↑ |
>     | CoLV              |   39.6   |  -   |  2.9 | 20.6 |  7.9 |    54.3    |  -   |  2.1 | 19.7 |  6.3 |
>     | KnowledGPT        |   19.2   |  9.5 |  7.2 | 22.0 |  7.9 |    22.3    |  8.3 |  6.0 | 20.5 |  6.7 |
>     | SPI               |   17.1   | 10.2 |  7.7 | 22.7 |  8.8 |    19.1    |  9.6 |  7.3 | 22.0 |  8.5 |
>     | R2C2-PT           |   24.7   |  6.0 |  4.4 | 17.3 |  5.2 |    30.4    |  6.2 |  4.6 | 17.7 |  5.4 |
>     | ELVM-from-R2C2-PT |   14.6   | 14.7 | 11.9 | 29.3 | 13.9 |    18.0    | 14.5 | 11.8 | 29.2 | 13.7 |
>     | R2C2              |   40.7   |  1.7 |  0.9 | 12.3 |  1.9 |    46.6    |  1.6 |  0.9 | 11.9 |  5.4 |
>     | ELVM-from-R2C2    |   15.8   | 14.5 | 11.8 | 28.8 | 13.3 |    18.9    | 14.3 | 11.5 | 28.8 | 13.3 |
>
> - **Conclusions:**
>
>     From the supplementary experiment, it is discernible that while R2C2-PT leads some advantages, the significant performance enhancements can be largely credited to our proposed algorithm, ELVM. This underscores ELVM's capability to drive performance improvements.
>
> **BART as Base:**
>
> We acknowledge the valid point raised about the potential unfair advantage due to the discrepancy between our initial R2C2 model (400M parameters) and the BART-base model (140M parameters) used for the previous models including the previous SOTA model (SPI) on the WoW task.
>
> - **Addressing the Model Size Discrepancy:**
>
>     To ensure a fair comparison and further demonstrate the robustness of our algorithm on diverse model scales, we conducted an experiment employing the BART-base as an initial model and then applied our proposed algorithm, ELVM, on the WoW dataset. The outcome of this training is termed “ELVM-from-BART.” Please note that due to the limted period and resource, we conduct this experiment on WoW dataset.
>
> - **Evaluation Results:**
>
>     The initial BART model shows very poor performances. However, upon training the BART-base with our ELVM algorithm, there is a great improvement in performance across both WoW Seen and Unseen tasks. Importantly, the results surpass the performance benchmarks set by the previous SOTA model (SPI). This endorses the robustness and scalability of our algorithm, confirming its efficacy irrespective of the model size, from BART-base's scale (140M) to a billion-scale parameter size (as elaborated in Section 4.4, line 539). We believe this addresses the primary concern regarding the potential advantage due to model size and underscores the genuine contributions of our research.
>
>     | Model             | WoW Seen |      |      |      |      | WoW Unseen |      |      |      |      |
>     |-------------------|:--------:|:----:|:----:|:----:|:----:|:----------:|:----:|:----:|:----:|:----:|
>     |                   |   PPL ↓  | B3 ↑ | B4 ↑ | R1 ↑ | R2 ↑ |    PPL ↓   | B3 ↑ | B4 ↑ | R1 ↑ | R2 ↑ |
>     | CoLV              |   39.6   |  -   |  2.9 | 20.6 |  7.9 |    54.3    |  -   |  2.1 | 19.7 |  6.3 |
>     | KnowledGPT        |   19.2   |  9.5 |  7.2 | 22.0 |  7.9 |    22.3    |  8.3 |  6.0 | 20.5 |  6.7 |
>     | SPI               |   17.1   | 10.2 |  7.7 | 22.7 |  8.8 |    19.1    |  9.6 |  7.3 | 22.0 |  8.5 |
>     | R2C2-PT           |   24.7   |  6.0 |  4.4 | 17.3 |  5.2 |    30.4    |  6.2 |  4.6 | 17.7 |  5.4 |
>     | ELVM-from-R2C2-PT |   14.6   | 14.7 | 11.9 | 29.3 | 13.9 |    18.0    | 14.5 | 11.8 | 29.2 | 13.7 |
>     | R2C2              |   40.7   |  1.7 |  0.9 | 12.3 |  1.9 |    46.6    |  1.6 |  0.9 | 11.9 |  5.4 |
>     | ELVM-from-R2C2    |   15.8   | 14.5 | 11.8 | 28.8 | 13.3 |    18.9    | 14.3 | 11.5 | 28.8 | 13.3 |
>     | BART              |   974.0  |  1.9 |  1.3 | 11.2 |  2.6 |    973.0   |  1.8 |  1.3 | 10.9 |  2.6 |
>     | ELVM-from-BART    |   15.3   | 10.6 |  8.4 | 25.8 | 11.1 |    16.4    | 10.4 |  8.2 | 25.7 | 10.9 |
>
> **Distinctions:**
>
> Variational latent modeling to knowledge-intensive dialogue tasks is not novel and used in previous works. Thereby, we agree that the terminology ‘novel’ is ambiguous and will be removed in the revised version. Instead, we will draw more clear illustration of the distinction of our method from previous works that applies variational techniques to knowledge-intensive dialogue tasks in the revised paper.
>
> In prior to the revision, we outline the distinctions between ELVM and [1] here.
> - First, the latent variables differ, as ELVM uses search queries as the latent variables, from which the relevant documents are retrieved using off-the-shelf retrievers, whereas [1] explicitly trains the retriever and treats the retrieved documents as the latent variable.
> - Second, ELVM is able to efficiently train the latent variable model by modifying the ELBO loss without extraneous encoder architectures or persistent update of document index.
>
> We furthermore demonstrate the state-of-the-art performances in knowledge-intensive benchmarks (WoW and QReCC), showing outperformances over previous approaches by a significant margin.

---

### Official Review · Reviewer_doRo · 2023-08-15

**Typos Grammar Style And Presentation Improvements:** + In table 1, improper citation for SKT.
**Soundness:** 3

**Excitement:**

2: Mediocre: This paper makes marginal contributions (vs non-contemporaneous work), so I would rather not see it in the conference.

**Paper Topic And Main Contributions:**

The paper proposes an efficient latent variable modeling algorithm, called ELVM, for knowledge-grounded dialogue generation. The algorithm leverages a large amount of dialogue data and enables joint training of the query generator and response generator without the need for annotated knowledge. The experimental results show that ELVM achieves state-of-the-art performances on diverse knowledge-grounded dialogue datasets.

**Questions For The Authors:**

+ Could you provide the upper bound performance figures for WoW and QReCC, assuming the golden knowledge is given?
+ What is the rationale behind employing a singular model for both the query generator and response generator? What if their parameters were not shared?

**Reasons To Accept:**

+ The paper introduces an innovative and efficient algorithm, ELVM, for knowledge-grounded dialogue generation.
+ The proposed algorithm leverages a large amount of dialogue data and achieves state-of-the-art performances.

**Reasons To Reject:**

+ The validity of the formula induction necessitates careful verification. Particularly, after the substitution of the posterior distribution $q(u \mid x,y)$, does the inequality in Equation 6 still maintain its validity? Additionally, could the disparity in the definitions of the posterior distribution $q(u \mid x,y)$ in Equations 8 and 9 potentially introduce convergence-related challenges?
+ The potential for an unfair comparison in the experiments should be considered. The backbone of baseline methods such as KnowGPT or CoLV differ from the proposed method, which is established based on R2C2. Moreover, these baseline models encompass highly distinct parameter scales. Consequently, unless the authors adapt the previous methods to employ the same backbone (i.e., R2C2), the comparison could lack meaningful interpretation.
+ A baseline method appears to be missing. ZRKGC[1] is a prominent technique in the context of knowledge-grounded dialogue, particularly in scenarios where knowledge annotations are unavailable.






[1] Zero-Resource Knowledge Grounded Dialogue Generation

**Reproducibility:**

3: Could reproduce the results with some difficulty. The settings of parameters are underspecified or subjectively determined; the training/evaluation data are not widely available.

**Reviewer Confidence:**

4: Quite sure. I tried to check the important points carefully. It's unlikely, though conceivable, that I missed something that should affect my ratings.

---

> ### Author Rebuttal · Authors · 2023-08-29
>
> We deeply appreciate the time and effort you have dedicated to reviewing our paper. Your insightful feedback has been instrumental in refining our work, and we are grateful for the constructive suggestions and the interest you've shown. We will incorporate your comments and feedback into our revised version. We have taken note of your concerns and questions and will address them in the sections below.
>
> **Validity of Equation 6:**
>
> It is possible that there is a bias that could be reduced using complex posterior modeling or posterior sampling (e.g., MCMC). Having said that, instead, for efficiency, we try to reduce the bias by sampling multiple queries. Namely, increasing the number of generated queries \( m \) aims to mitigate the bias caused by sampling from the prior. The results in Table 5 show improved performances as \( m \) increases, and a plateau upon reaching \( m = 16 \). We will apply and compare the use of posterior modeling and posterior sampling in the revised paper.
>
> **Posterior Distribution in Equation 9:**
>
> Please note that the stop gradient is set on the posterior distribution, denoted as \( q(u|x,y) \) in Equation 9, which means that the common network trained by both equation 8 and 9 is only the prior network, and the prior network is updated to catch the posterior network in Equation 9. Since both loss functions are applied to update the same prior network, we did not encounter any instability or convergence issue during ELVM's training.
>
> **Use of R2C2 in ELVM's Training:**
> - **Addressing the Central Concern:**
>
>     We deeply appreciate the thorough feedback regarding the possible attribution of ELVM's performance improvements to the utilization of R2C2-PT.
>
>     To rigorously address the aforementioned concern, we have undertaken an additional experiment, deliberately omitting the post-pretraining stage. Our focus was to empirically validate whether the gains in performance arise primarily from the utilization of R2C2-PT or if they can be ascribed to the efficacy of our proposed algorithm, ELVM.
>
>
> - **Experiment Design:**
>
>     We start with a R2C2 transformer model as our base rather than R2C2-PT and then train with ELVM algorithm. The inference settings remained unaltered from Section 4.1 (line 422).
>
>     To avoid ambiguity, we name this resulting model as “ELVM-from-R2C2” and rename the model trained from R2C2-PT as "ELVM-from-R2C2-PT", which corresponds to the "ELVM" terminology utilized in our original submission.
>
> - **Results:**
>
>     Our results illustrate that while the standalone R2C2 model showed poor performances, our proposed ELVM leads to great improvements (ELVM-from-R2C2). Note that this improved performance is better than that of the previous SOTA model (SPI) and is comparable to the result obtained by ELVM-from-R2C2-PT.
>
>     | Model             | WoW Seen |      |      |      |      | WoW Unseen |      |      |      |      |
>     |-------------------|:--------:|:----:|:----:|:----:|:----:|:----------:|:----:|:----:|:----:|:----:|
>     |                   |   PPL ↓  | B3 ↑ | B4 ↑ | R1 ↑ | R2 ↑ |    PPL ↓   | B3 ↑ | B4 ↑ | R1 ↑ | R2 ↑ |
>     | CoLV              |   39.6   |  -   |  2.9 | 20.6 |  7.9 |    54.3    |  -   |  2.1 | 19.7 |  6.3 |
>     | KnowledGPT        |   19.2   |  9.5 |  7.2 | 22.0 |  7.9 |    22.3    |  8.3 |  6.0 | 20.5 |  6.7 |
>     | SPI               |   17.1   | 10.2 |  7.7 | 22.7 |  8.8 |    19.1    |  9.6 |  7.3 | 22.0 |  8.5 |
>     | R2C2-PT           |   24.7   |  6.0 |  4.4 | 17.3 |  5.2 |    30.4    |  6.2 |  4.6 | 17.7 |  5.4 |
>     | ELVM-from-R2C2-PT |   14.6   | 14.7 | 11.9 | 29.3 | 13.9 |    18.0    | 14.5 | 11.8 | 29.2 | 13.7 |
>     | R2C2              |   40.7   |  1.7 |  0.9 | 12.3 |  1.9 |    46.6    |  1.6 |  0.9 | 11.9 |  5.4 |
>     | ELVM-from-R2C2    |   15.8   | 14.5 | 11.8 | 28.8 | 13.3 |    18.9    | 14.3 | 11.5 | 28.8 | 13.3 |
>
> - **Conclusions:**
>
>     From the supplementary experiment, it is discernible that while R2C2-PT leads some advantages, the significant performance enhancements can be largely credited to our proposed algorithm, ELVM. This underscores ELVM's capability to drive performance improvements.
>
> **BART as Base:**
>
> We acknowledge the valid point raised about the potential unfair advantage due to the discrepancy between our initial R2C2 model (400M parameters) and the BART-base model (140M parameters) used for the previous models including the previous SOTA model (SPI) on the WoW task.
>
> - **Addressing the Model Size Discrepancy:**
>
>     To ensure a fair comparison and further demonstrate the robustness of our algorithm on diverse model scales, we conducted an experiment employing the BART-base as an initial model and then applied our proposed algorithm, ELVM, on the WoW dataset. The outcome of this training is termed “ELVM-from-BART.” Please note that due to the limted period and resource, we conduct this experiment on WoW dataset.
>
> - **Evaluation Results:**
>
>     The initial BART model shows very poor performances. However, upon training the BART-base with our ELVM algorithm, there is a great improvement in performance across both WoW Seen and Unseen tasks. Importantly, the results surpass the performance benchmarks set by the previous SOTA model (SPI). This endorses the robustness and scalability of our algorithm, confirming its efficacy irrespective of the model size, from BART-base's scale (140M) to a billion-scale parameter size (as elaborated in Section 4.4, line 539). We believe this addresses the primary concern regarding the potential advantage due to model size and underscores the genuine contributions of our research.
>
>     | Model             | WoW Seen |      |      |      |      | WoW Unseen |      |      |      |      |
>     |-------------------|:--------:|:----:|:----:|:----:|:----:|:----------:|:----:|:----:|:----:|:----:|
>     |                   |   PPL ↓  | B3 ↑ | B4 ↑ | R1 ↑ | R2 ↑ |    PPL ↓   | B3 ↑ | B4 ↑ | R1 ↑ | R2 ↑ |
>     | CoLV              |   39.6   |  -   |  2.9 | 20.6 |  7.9 |    54.3    |  -   |  2.1 | 19.7 |  6.3 |
>     | KnowledGPT        |   19.2   |  9.5 |  7.2 | 22.0 |  7.9 |    22.3    |  8.3 |  6.0 | 20.5 |  6.7 |
>     | SPI               |   17.1   | 10.2 |  7.7 | 22.7 |  8.8 |    19.1    |  9.6 |  7.3 | 22.0 |  8.5 |
>     | R2C2-PT           |   24.7   |  6.0 |  4.4 | 17.3 |  5.2 |    30.4    |  6.2 |  4.6 | 17.7 |  5.4 |
>     | ELVM-from-R2C2-PT |   14.6   | 14.7 | 11.9 | 29.3 | 13.9 |    18.0    | 14.5 | 11.8 | 29.2 | 13.7 |
>     | R2C2              |   40.7   |  1.7 |  0.9 | 12.3 |  1.9 |    46.6    |  1.6 |  0.9 | 11.9 |  5.4 |
>     | ELVM-from-R2C2    |   15.8   | 14.5 | 11.8 | 28.8 | 13.3 |    18.9    | 14.3 | 11.5 | 28.8 | 13.3 |
>     | BART              |   974.0  |  1.9 |  1.3 | 11.2 |  2.6 |    973.0   |  1.8 |  1.3 | 10.9 |  2.6 |
>     | ELVM-from-BART    |   15.3   | 10.6 |  8.4 | 25.8 | 11.1 |    16.4    | 10.4 |  8.2 | 25.7 | 10.9 |
>
> **Comparison with ZRKGC:**
>
> In the revised version, we will incorporate the experiments for ZRKGC, complete with appropriate citations. For note, a pivotal distinction (Section 2.2) between our approach and ZRKGC is in the query generation methodology: our model explicitly generates search queries to retrieve relevant documents, while ZRKGC selects pertinent documents based on a similarity metric between the ground truth response and the documents. Thereby, the latent variable is search query for ELVM whereas it is document and knowledge grounding rate in ZRKGC.
>
> Furthermore, as discussed towards the end of Section 3.2, our proposed method differs from EM algorithm in that the expectation calculated using sampled from the prior instead of the posterior, through which the learning gradient flows hence training the prior accordingly. Moreover, there is a substantial performance discrepancy between our model and ZRKGC on the WoW test set. The performance metrics for the WoW Seen and Unseen tasks are delineated below:
>
> | Model | WoW Seen |      | WoW Unseen |      |
> |-------|:--------:|------|:----------:|------|
> |       |   PPL ↓  | F1 ↑ |    PPL ↓   | F1 ↑ |
> | ZRKGC |   40.4   | 18.7 |    41.5    | 18.6 |
> | ELVM  |   14.6   | 27.2 |    14.5    | 27.0 |
>
> ---
>
> **Question 1. Upper Bound**
>
> We believe in the value of understanding the utmost capability in an ideal scenario. To this end, we pursued an upper-bound performance analysis for both the WoW and QReCC tasks. This approach involves training our response generator under the assumption that the model has access to the ground-truth document during both training and evaluation. The performance metrics of this specific setup are designated as "ELVM-w-GT-Doc" in the subsequent table.
>
> - WoW Result
>     | Model         | WoW Seen |      |      |      |      | WoW Unseen |      |      |      |      |
>     |---------------|:--------:|:----:|:----:|:----:|:----:|:----------:|:----:|:----:|:----:|------|
>     |               |   PPL ↓  | B3 ↑ | B4 ↑ | R1 ↑ | R2 ↑ |    PPL ↓   | B3 ↑ | B4 ↑ | R1 ↑ | R2 ↑ |
>     | ELVM          |   14.6   | 14.7 | 11.9 | 29.3 | 13.9 |    18.0    | 14.5 | 11.8 | 29.2 | 13.7 |
>     | ELVM-w-GT-Doc |    9.7   | 22.1 | 18.5 | 40.9 | 23.9 |    10.5    | 21.5 | 17.9 | 40.7 | 23.3 |
>
> - QReCC Result
>     |     Model     |  F1↑  |  EM↑  |
>     |:-------------:|:----:|:----:|
>     | ELVM          | 36.5 |  6.2 |
>     | ELVM-w-GT-Doc | 55.4 | 16.2 |
>
> From the table, it's evident that there still remains a margin for improvement. We are committed to integrating this profound insight into our revision, as it not only showcases the potential of our approach but also underlines the extensive future work for more improvement.
>
> **Question 2. Parameter Share**
>
> In response to the question regarding parameter sharing in the training of ELVM, we conducted an additional experiment wherein we decoupled weights between the query generator and the response generator, leading to the creation of the "EVLM-Decouple" model. We evaluate the performance of EVLM-Decouple on both WoW test seen and unseen tasks and report in the table below.
>
> | Model         | WoW Seen |      |      |      |      | WoW Unseen |      |      |      |      |
> |---------------|:--------:|:----:|:----:|:----:|:----:|:----------:|:----:|:----:|:----:|:----:|
> |               |   PPL ↓  | B3 ↑ | B4 ↑ | R1 ↑ | R2 ↑ |    PPL ↓   | B3 ↑ | B4 ↑ | R1 ↑ | R2 ↑ |
> | ELVM          |   14.6   | 14.7 | 11.9 | 29.3 | 13.9 |    18.0    | 14.5 | 11.8 | 29.2 | 13.7 |
> | ELVM-Decouple |   16.2   | 15.5 | 12.6 | 30.1 | 14.5 |    21.6    | 15.0 | 12.2 | 29.7 | 14.0 |
>
> Upon analysis of the results, we note a marginal performance improvement by EVLM-Decouple in comparison to ELVM. It is, however, essential to highlight the intrinsic advantage of the shared model approach, specifically in terms of computational resource efficiency and knowledge transfer during both the training and inference phases.
>
> We acknowledge your feedback and will certainly incorporate this experimental result into the revised version of the paper.

---

### Meta-Review · Area_Chair_8PA5 · 2023-09-12

**Recommendation:** 2

**Metareview:**

This paper introduces a new approach to knowledge-grounded dialogue generation, termed Efficient Latent Variable Modeling (ELVM). ELVM incorporates a query generator, followed by an off-the-shelf retriever, which remains fixed during the training process. Notably, ELVM aims to facilitate efficient and robust training without the requirement for annotated knowledge or explicit training of knowledge retrieval.

While the utilization of variational techniques to enhance knowledge-intensive text generation tasks is not new, ELVM demonstrates its effectiveness in unsupervised joint training, leading to substantial performance enhancements compared to prior methods. However, despite the novelty of ELVM, the actual improvements over existing methods may appear relatively modest.

---

### Decision · Program_Chairs · 2023-10-07

**Decision:**

Accept-Findings

**Comment:**

This paper introduces a new approach to knowledge-grounded dialogue generation, termed Efficient Latent Variable Modeling (ELVM). ELVM incorporates a query generator, followed by an off-the-shelf retriever, which remains fixed during the training process. Notably, ELVM aims to facilitate efficient and robust training without the requirement for annotated knowledge or explicit training of knowledge retrieval.

While the utilization of variational techniques to enhance knowledge-intensive text generation tasks is not new, ELVM demonstrates its effectiveness in unsupervised joint training, leading to substantial performance enhancements compared to prior methods. However, despite the novelty of ELVM, the actual improvements over existing methods may appear relatively modest.